# Shared fate was associated with sustained cooperation during the COVID-19 pandemic

**Diego Guevara Beltran** [1,2‡]*, **Jessica D. Ayers** [2,3‡], **Scott Claessens** [2,4], **Joe Alcock** [2,5], **Cristina Baciu** [2,6], **Lee Cronk** [2,7], **Nicole M. Hudson** [2,6], **Hector Hurmuz-Sklias** [2], **Geoffrey Miller** [2,8], **Keith Tidball** [2,9], **Andrew Van Horn** [2,10], **Pamela Winfrey** [2,6], **Emily Zarka** [2,11], **Peter M. Todd** [2,12‡], **Athena Aktipis** [2,6,13‡]

1 Department of Psychology, University of Arizona, Tucson, Arizona, United States of America, 2 Zombie Apocalypse Medicine Alliance, Tempe, Arizona, United States of America, 3 Department of Psychological Science, Boise State University, Boise, Idaho, United States of America, 4 School of Psychology, University of Auckland, Auckland, New Zealand, 5 Department of Emergency Medicine, University of New Mexico, Albuquerque, New Mexico, United States of America, 6 Department of Psychology, Arizona State University, Tempe, Arizona, United States of America, 7 Department of Anthropology, Rutgers University, New Brunswick, New Jersey, United States of America, 8 Psychology Department, University of New Mexico, Albuquerque, New Mexico, United States of America, 9 Department of Natural Resources and Environment, Cornell University, Ithaca, New York, United States of America, 10 Department of Physics and Art History, Case Western Reserve University, Cleveland, Ohio, United States of America, 11 Department of English, Arizona State University, Tempe, Arizona, United States of America, 12 Cognitive Science Program and Department of Psychological and Brain Sciences, Indiana University, Bloomington, Indiana, United States of America, 13 Biodesign Center for Biocomputing, Security and Society, Arizona State University, Tempe, Arizona, United States of America

‡ DGB and JDA are share first authorship on this work. AA and PMT are share senior authorship on this work.
* dguevarabeltran@arizona.edu

**Data Availability Statement:** Data is available here: https://osf.io/gtsbr/.

## Abstract

Did the COVID-19 pandemic bring people together or push them apart? While infectious diseases tend to push people apart, crises can also bring people together through positive interdependence. We studied this question by asking an international sample (N = 1,006) about their inclinations to cooperate, perceptions of interdependence (i.e., shared fate), and perceived risk as well as local prevalence of COVID-19 infection across 14 time points from March to August, 2020. While perceived interdependence with others tended to increase during this time period, inclinations to cooperate decreased over time. At the within-person level, higher local prevalence of COVID-19 attenuated increases in perceived interdependence with others, and was associated with lower inclinations to cooperate. At the between-person level, people with high perceived interdependence with others reported more stable, or increasing, inclinations to cooperate over time than people with low perceived interdependence. Establishing a high sense of perceived interdependence with others may thus allow people to maintain cooperation during crises, even in the face of challenging circumstances such as those posed by a highly transmissible virus.

**Funding:** This study was funded by the Interdisciplinary Cooperation Initiative, the President's Office of Arizona State University, the Cooperation Science Network, the Institute for Mental Health Research, the University of New Mexico, the Indiana University College of Arts & Sciences, the Rutgers University Center for Human Evolutionary Studies, the Charles Koch Foundation, the John Templeton Foundation, and the President's Office of the University of Arizona. Any opinions, findings, conclusions or recommendations expressed in this material are those of the authors and do not necessarily reflect the views of the funding entities mentioned above.

**Competing interests:** We have no conflicts of interest to disclose.

## Introduction

Times of crisis can bring out the best and worst sides of human nature. Sometimes people become fearful and aggressive, stealing resources and committing crimes [1]; other times they show "catastrophe compassion," providing help to those in need [1, 2]. Crises can create positive interdependence, eliciting perceptions such as shared fate and shared identity and, in turn, promoting cooperation [3, 4]. However, the COVID-19 pandemic might have also created negative interdependence by way of disease transmission. In contrast to shared fate, infection risk tends to induce avoidance, reducing shared fate and inhibiting cooperation [5–8]. Hence, not only did the COVID-19 pandemic evoke the duality of human nature observed during previous crises, but people also had to weigh their inclinations to cooperate against an increasing risk of infection [9–11].

Given these competing observations, our primary aim was to test how the prevalence of COVID-19, perceived risk of infection, and perceived interdependence influenced people's inclinations to cooperate across the first six months of the pandemic. Moreover, because of the unique challenges posed by the COVID-19 pandemic relative to previously studied crises (in particular, increasing risk of infection), we first provide a descriptive account of the developmental trajectory of perceived interdependence and inclinations to cooperate across the first six months of the pandemic.

## People engaged in cooperation and conflict during the COVID-19 pandemic

During the pandemic, volunteers distributed food to needy families, people volunteered in testing facilities, military personnel operated testing sites and built hospitals, and healthy volunteers were infected with COVID-19 for a human-challenge trial to speed up vaccine development [12, 13]. Others helped by hosting family members during lockdowns, working extra shifts to keep the food supply chains running, checking on vulnerable neighbors [14, 15], or contributing to mutual-aid organizations across the United States [16, 17]. All of this happened even though contact with others increased helpers' risk of contracting the very infection they were trying to combat.

At the same time, more people were purchasing household goods (e.g., cleaning supplies, non-perishable foods) during March 2020 compared to the previous year [18], fueling the belief that people were looking out for themselves (e.g., hoarding) and incentivizing others to behave similarly [19]. Such fears, while factually unfounded [18], even led the Secretary of Health and Human Services of the United States to take action to mitigate the potential risk of hoarding and price gouging [20]. Moreover, the COVID-19 pandemic may have also hindered cooperation by way of outgroup prejudice. For example, preexisting Islamophobia was exacerbated in India during the pandemic [21], and discrimination towards Asian people became more prevalent because some assumed Asian individuals were more likely to be vectors of COVID-19 [10, 22–24]. We do not mean to imply that people are either selfish or cooperative but rather that both behaviors could be observed, as is often the case during times of crises [1]. In addition, given the high levels of uncertainty posed by a novel pandemic, people's cooperative or selfish behaviors were largely influenced by locally relevant descriptive norms [25].

### Conflict and cooperation are shaped by fitness interdependence

Why do people sometimes cooperate and other times engage in conflict? Fitness interdependence theories predict that individuals cooperate when their outcomes are correlated in ways that benefit each other's survival and reproduction but engage in conflict when their outcomes

are correlated in ways that hinder each other's survival and reproduction [26–30]. Many different settings can create a positive stake in another's survival and reproduction (i.e., positive fitness interdependence), giving rise to perceptions of positive interdependence (e.g., perceived similarity, shared identity/oneness, shared fate). These settings can include genetic relatedness, sharing resources, mutual aid, group membership, and intergroup conflict [31]. When people perceive high shared fate in their relationships – the belief that their outcomes are positively intertwined – they value such relationships more highly and are more likely to help one another in times of need [32–35].

## Settings of fitness interdependence and cues associated with perceived interdependence

Settings of fitness interdependence are the contexts in which people's outcomes can become positively (or negatively) correlated, and they can have different cues associated with them. For example, because co-residence (setting) during childhood and true siblingship are tightly coupled, growing up together (cue) is a reliable indicator of positive fitness interdependence by way of genetic relatedness. Hence, siblings who grew up together are more likely to help one another than siblings who did not grow up together [36, 37]. Another setting of fitness interdependence is the sharing of resources and mutual aid. For example, in food- or labor-sharing networks (setting), a history of sharing (cue) is a reliable indicator of fitness interdependence by way of pooling calories and other sources of risk. Hence, those who share are more likely to receive help when in need [38–42]. A third setting of fitness interdependence is group membership (e.g., ethnic, religious). Groups can control the access and distribution of resources. As a result, behaviors that signal group membership (setting), such as participation in costly rituals (cue), are reliable indicators of fitness interdependence by way of group membership. And, hence, people who participate in religious rituals receive more help from other members of their group than people who do not participate in these rituals [43, 44].

## Crises as settings of fitness interdependence

Crises can yoke together people's outcomes. For example, when people experience fires with an interdependent partner (e.g., friend, spouse), they are 55% more likely to die if their partner passes away than people who have no ties to others in the fire [45]. This outcome is likely due to people escaping burning buildings in groups or dyads when they share close ties with others, but individually when they do not share a close tie with another person in the fire [46].

Crises can increase perceptions of self-other merging, mutual support, and group membership, which can increase perceived shared fate and cooperation [3, 4, 47, 48]. Nine-year-olds who experienced greater exposure to an earthquake were more likely to perceive other children as ingroup members, feel closer with other children, and report greater intentions of helping children who experienced the earthquake [47, 48]. Among Chilean and Italian adults, people who were more negatively affected by an earthquake reported greater perceptions of shared fate and oneness and greater rates of helping others in need [3, 4].

Feelings of shared fate are also often enhanced in intergroup conflict settings. For example, after the Boston marathon bombing in the US, people reported higher kinship and self-other merging with members of the US, as well as higher willingness to help those affected by the bombing [49, 50]. Following the terrorist attacks of September 11[th] in the US, 30-65% of people reported helping those affected by the attack (e.g., donating blood, monetary donations) [51–54]. When people engage in combat together, they often report greater self-other merging with one another than with their close kin [55, 56].

## Infection risk as a source of negative interdependence

Exposure to a collective threat and experiencing opportunities to engage in mutual support are features shared by previous crises and the COVID-19 pandemic. However, unlike previously studied crises, the pandemic brought additional challenges associated with the transmissibility of a highly infectious virus. Moreover, unlike other crises in which the risks are localized in time and space (e.g., earthquakes, floods, terrorist attacks), the COVID-19 pandemic continues to be a global and dynamic crisis, initially characterized by high levels of uncertainty and stress-inducing events [57, 58]. The novelty and uncertainty surrounding the pandemic also contributed to a lack of clear guidelines for collective behavior (e.g., mask wearing) during the early periods of the pandemic [59]. And even when clear guidelines for collective behavior became widespread, adherence to such guidelines varied widely [25, 60–62]. The transmissibility of the disease and general uncertainty surrounding the pandemic posed a challenge to perceiving positive interdependence and engaging in cooperation because the risk of infection and the risk of defection leads to the possibility of negative fitness interdependence (e.g., becoming infected when helping a person in need, helping an exploitative person).

The ability to detect dangers from potentially infectious conspecifics, feelings of disgust, and resultant distancing behaviors are components of what is termed the behavioral immune system. This set of cognitive, affective, and behavioral mechanisms are thought to mobilize individuals to reduce the likelihood of contracting diseases [6–8, 63]. When the behavioral immune system is activated (e.g., by perceived risk of infection or feelings of disgust) people perceive lower shared fate with others [5] and become less willing to cooperate [64, 65] or even interact with potentially infectious others, especially outgroup members [66, 67]. These findings are in line with the idea that people are more willing to engage in behaviors that put them at risk of infection with people they value than with those they do not value as highly [68].

During the COVID-19 pandemic, disease-avoidance concerns were expectedly higher across the world compared to before the pandemic [69]. This led researchers to ask whether changes brought upon by the pandemic (i.e., exposure to disease) resulted in increased outgroup rejection. However, results from longitudinal studies have been mixed. One set of findings suggest that changes brought upon by the pandemic (e.g., individual exposure to disease-related cues) did not lead to changes in outgroup rejection, but rather that pre-existing individual differences predicted outgroup rejection among people high on trait-disease avoidance concerns [70, 71]. In contrast, another study shows that individual differences in disease avoidance motivation and situational changes in disease avoidance motivation were associated with a higher motivation to avoid potentially infectious targets (i.e., US Republicans) for both ingroups (i.e., Republicans) and outgroups (i.e., Democrats) [72].

## How do crises influence perceived interdependence and cooperation over time?

Disasters, like other settings (e.g., intergroup conflict), can be thought of as a setting of fitness interdependence *in so far as* the disaster generates conditions or situations in which people's outcomes become correlated, and this fitness interdependence in turn elicits perceptions of interdependence (e.g., self-other-merging with one's group, shared fate). To understand whether risk of infection and shared fate influenced cooperation over time during the COVID-19 pandemic, we first need to understand how these constructs unfold over time during crises. While we recognize that pandemics differ meaningfully from other crises (e.g., transmissibility of disease, localization, duration), these previous studies can provide general clues as to how people might respond over time. Some research suggests there are increases in cooperation in the immediate aftermath of a crisis. Five days after the Boston marathon

bombing in the US, people who reported higher self-other merging with other people in the US were more likely to donate money to those affected by the attack [49]. People who engaged in demonstrations against the terrorist attacks of March 11, 2004, in Spain reported having greater social support three weeks after the attack [73]. Similarly, in the US, volunteer rates increased two-fold during the two-week period following the terrorist attacks of September 11, 2001. However, volunteer rates returned to baseline levels after two weeks [74].

In parallel to volunteer rates following terrorist attacks, people who experienced greater losses during a flood reported receiving more help shortly after the flood, but perceptions of received support declined considerably six months after the flood [75, 76]. This decline in perceived support following crises is a result of unmet needs or expectations, rather than declines in actual received support per se. Controlling for received support, people who experienced more losses during a hurricane reported lower perceptions of received support two years after the hurricane, suggesting that unmet needs contribute to declines in perceived support regardless of objectively received support [77]. Moreover, greater exposure to disasters can lead to or exacerbate mental health problems (e.g., PTSD) which, in the longer term (i.e., 12-18 months post disaster), can lead to steeper declines in perceptions of received support [78].

In a meta-analysis on intergroup conflict and cooperation, exposure to violence was positively associated with cooperation in economic games. Studies ranged from a few months to 12 years post-conflict, indicating that exposure to violence in times of war is associated with persistent increases in cooperation. However, these increases in cooperation were only observed when people had the opportunity to cooperate with members of their own group, but not when interacting with outgroup members [79]. This effect of increased cooperation with ingroup members but not outgroup members can be attributed to ingroup members being more likely to cooperate (i.e., act as potential sources of positive interdependence), while outgroup members are less likely to cooperate (i.e., act as potential sources of negative interdependence). During the COVID-19 pandemic, we might also see increases in cooperation, but only towards ingroup members.

Although few previous studies have investigated how perceived interdependence changes over time following a crisis, some have investigated changes in relationship strength (e.g., closeness, belongingness). Measures of relationship strength such as closeness (or self-other-overlap) are strongly positively correlated with shared fate [32, 35]. Hence, these previous studies can provide clues as to how shared fate might change over time in times of crises. Following the Madrid terrorist attacks of March 11, 2004, people reported small increases in perceived belongingness with their friends that lasted up to eight weeks after the attacks [80]. After a disaster (e.g., flood, earthquake), people who were more negatively impacted (e.g., lack of resources) reported small to medium positive growth (e.g., closeness) in their overall relationships with others that could still be seen up to four years later [81, 82]. Similarly, adolescents who engaged in mutual support after a flood reported stronger relationships 21 months after the flood [83]. While these studies did not measure relationship strength over many time periods, these difference scores suggest small to medium positive increases in relationship strength following a crisis.

Studies during the COVID-19 pandemic point to a pattern of initial increases in cooperation followed by decreases in the longer term. For example, activity in online mutual aid groups (i.e., posts requesting and offering help) increased sharply during the first wave of the pandemic (i.e., March 2020), but declined to baseline levels by July 2020 [84]. Among Spanish adults, people who socially isolated with others during the first week of the pandemic reported greater self-other merging with others, in turn leading to greater prosocial compliance with norms [85]. While a second study also found that self-other merging with one's country was associated with higher inclinations to cooperate in March, 2020, self-other merging and

inclinations to cooperate were lower by August, 2020 [86]. These studies suggest that perceived interdependence and inclinations to cooperate decreased over the first six months of the COVID-19 pandemic after an initial increase. However, in a nationally representative sample of US adults, cooperation (measured as generosity in the dictator game) was higher across the first six months of the pandemic during occasions in which there was a greater prevalence of COVID-19 threat in people's local environment [87].

## The present study

The pandemic created a unique time of collective need, providing an opportunity to investigate how perceptions of interdependence might influence inclinations to cooperate even during a time in which infection risk might push people to avoid helping others. Given the competing motivational outcomes of infection risk and perceived interdependence on cooperation, the primary aim of this study was to test whether the risk of infection and perceived interdependence changed people's inclinations to cooperate over the first six months of the pandemic. Moreover, because previous studies have not provided a descriptive account of how perceived interdependence changes over time during crises, nor of how cooperation changes over time during a pandemic, we first provide a detailed descriptive account (measured over 14 time points) of the developmental trajectory of these two constructs.

We predicted that perceived interdependence would be associated with greater cooperation, while risk of infection would be associated with lower cooperation. Moreover, we expected cooperation and perceived interdependence to be higher towards ingroup than towards outgroup members, as has been consistently shown in previous studies [79]. However, because moderating effects will depend on the shape of the trajectory of cooperation (i.e., whether cooperation is marginally stable, decreasing, or increasing over time), we did not have specific predictions about how risk of infection and perceived interdependence would moderate the change in cooperation over time.

For example, cooperation could marginally increase over time, as has been reported in some crises. If so, we might see that in times in which people experience high infection risk, they will report lower inclinations to cooperate (i.e., a within-person effect). However, we might also see that cooperation increases for those who experience chronically low infection risk but not those who experience chronically high infection risk (i.e., a between-person effect). Similarly, we might see that in times where people perceive high interdependence with others they will report higher inclinations to cooperate (i.e., a within-person effect). However, we might also see that cooperation increases among those who perceive high and stable interdependence with others, but not for those who perceive low and stable interdependence with others (i.e., a between-person effect).

One might also predict that perceived interdependence would marginally increase over time because related constructs (e.g., closeness) have been reported to increase during previous crises [80, 83]. However, because infection risk is associated with lower perceived interdependence, we might see that when people experience high infection risk they will report lower perceived interdependence with others (i.e., a within-person effect). We might also see that perceived interdependence with others only increases for people who experience chronically low, but not chronically high, infection risk (i.e., a between-person effect).

We successfully assessed these predictions by implementing both a person-centered and a trait-level approach. Specifically, we examined whether exposure to the COVID-19 pandemic influenced perceptions of interdependence and risk of infection, leading to changes in cooperation over time, or whether pre-existing individual differences in these measures accounted for changes in cooperation during the pandemic. Using a person-centered approach, we found

**Table 1. Sample size, attrition, and study duration per time period.**

|  | T1 | T2 | T3 | T4 | T5 | T6 | T7 | T8 | T9 | T10 | T11 | T12 | T13 | T14 |
|---|---|---|---|---|---|---|---|---|---|---|---|---|---|---|
| **Month** | Mar | Mar | Mar | Apr | Apr | May | May | May Jun | Jun | Jun Jul | Jul | Jul | Aug | Aug |
| **Days** | 6-7 | 14-17 | 24-27 | 3-6 | 17-22 | 2-7 | 16-21 | 30-3 | 13-17 | 27-1 | 11-15 | 25-29 | 8-12 | 22-26 |
| *N* | 505 | 403 | 892 | 834 | 802 | 692 | 752 | 736 | 708 | 701 | 745 | 686 | 628 | 668 |
| **Attrition (%)** | - | 20.2* | 23.3* | 17.1 | 20.3 | 31.2 | 25.2 | 26.8 | 29.6 | 30.3 | 25.9 | 31.8 | 37.6 | 33.6 |
| $M_{minutes}$ | 16.6 | 13.1 | 9.9 | 6.8 | 12.9 | 11.7 | 12.6 | 10.6 | 17.9 | 13.1 | 6.7 | 14.9 | 14.4 | 7.9 |

*Note*. Number of participants shows the effective sample size.

* = attrition rates for cohort 1 at time 2 and time 3. Attrition rates for times 4-14 are based on the total sample of 1006 participants. Time taken to complete the study differed between time points because some survey items intended for the purposes of different studies varied across time points.

that infection risk compromised cooperation and interfered with positive perceived interdependence. Using the trait-level approach, we found that people who perceived high interdependence with others reported higher and more stable inclinations to cooperate than people who perceived low interdependence with others.

## Method

### Participants and procedure

We recruited two cohorts through Prolific.co ($N_{Total}$ = 1018). At time 1 (March 6, 2020), we recruited 512 participants, and at time 3 (March 24, 2020) an additional 506 participants joined the study. We removed 12 participants who indicated they were not fluent in English, yielding an effective sample size of 1006 participants ($M_{age}$ = 28.65, $SD_{age}$ = 10.27, $Min_{age}$ = 18, $Max_{age}$ = 80, 50.35% men). People were initially invited to complete the survey every 10 days during a 72-hour period. Beginning at time 5 (April 17, 2020), participants were invited to complete the study every 15 days during a 96-hour period. Participants were compensated with $1.75 USD for times 1-4 ($10.10 USD/hour on average), and an average of $1.50 USD ($8.64 USD/hour on average) for times 5-14. See Table 1 for dates, sample size, and attrition rates.

Before beginning the study, participants read the electronic informed consent section. Next, they were directed to a page in which they were asked whether they agreed to participate in the study. Participants were instructed to click "continue" if they agreed to participate. Neither written nor oral consent forms were collected. All participants consented by choosing to continue/participate in the study. This study was deemed exempt and approved by the Institutional Review Board at Arizona State University. Demographic information was collected during each cohort's first survey. Most participants identified as White/Caucasian (80.6%), followed by Hispanic/Latino/a (6.7%), Asian/Pacific Islander (6.7%), "Other" (2.6%), Black/African American (1.7%), and Native/Native American (0.2%). Most participants resided in continental Europe (51.4%), followed by the UK/Ireland (28.1%), and North America (17.1%) (Fig 1; Table S1 in S1 File shows number of participants per country).

### Measures

Given the range of possible cues that might reflect perceived interdependence, as well as the range of possible behaviors that can reflect cooperation, we specifically employed tools that measured global perceived interdependence and global inclinations to cooperate with two types of targets that differed in ingroup/outgroup status (i.e., neighbors and all of humanity).

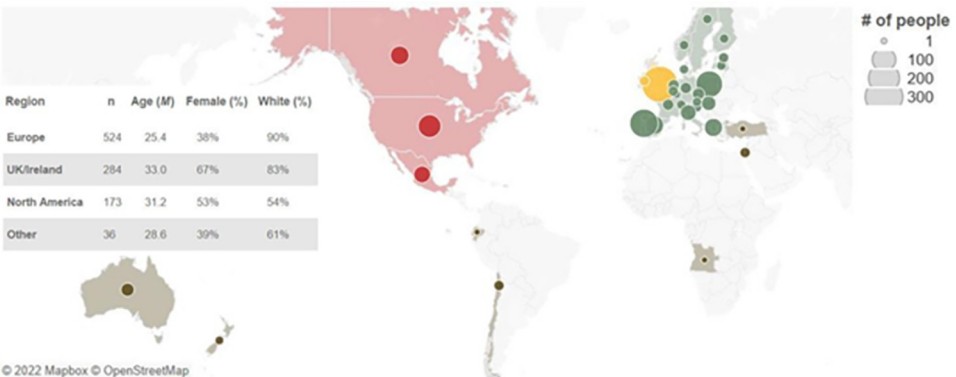

| Region | n | Age (M) | Female (%) | White (%) |
|---|---|---|---|---|
| Europe | 524 | 25.4 | 38% | 90% |
| UK/Ireland | 284 | 33.0 | 67% | 83% |
| North America | 173 | 31.2 | 53% | 54% |
| Other | 36 | 28.6 | 39% | 61% |

**Fig 1. Sample characteristics.** The size of circles shows the number of respondents. We created Fig 1 with OpenStreetMap® (https://www.openstreetmap.org). OpenStreetMap is open data, licensed under the Open Data Commons Open Database License (ODbL) by the OpenStreetMap Foundation (OSMF). According to OpenStreetMap "You are free to copy, distribute, transmit and adapt our data, as long as you credit OpenStreetMap and its contributors".

We measured cooperation toward neighbors and humanity via reported willingness to help (*Someone from your neighborhood* [*A person who is not a citizen of your own country*] *is having their residence fixed, so it isn't livable. How willing would you be to let them move into your residence for a week*?; 1 = *not at all willing*, 7 = *very willing)* and need-based helping attitudes (*Helping someone from my neighborhood* [*a person who is not a citizen of your own country*] *when they are in need is the right thing to do*; 1 = *strongly disagree*, 7 = *strongly agree*).

Participants then reported their perceived interdependence toward neighbors and humanity as indexed by perceived shared fate (*My neighborhood* [*All of humanity*] *and I rise and fall together*), and emotional shared fate (*When my neighborhood* [*all of humanity*] *succeeds, I feel good*; 1 = *do not agree at all*, 7 = *strongly agree*). Depending on the setting and source, fitness interdependence can be manifested psychologically through perceptions of interdependence such as perceived similarity in a relatedness or kin relationships setting, via self-other merging in an intergroup or large-scale disasters setting; and relationship closeness in a romantic relationships setting. At their core, these psychological manifestations of fitness interdependence track the extent to which others' outcomes are likely to influence one's own outcomes. The shared fate scale was designed to measure perceptions of interdependence globally (i.e., across settings, sources, and relationships), and is a reliable measure comprising two subscales that have shown concurrent, discriminant, convergent, and predictive validity [32].

Perceived shared fate tracks beliefs about the extent to which a partner's rewarding and aversive outcomes are likely to translate into a personal reward or loss– for example, the extent to which people believe a partner's job promotion will have a positive influence on their own life, or the extent to which people believe a partner's parent's death will have a negative influence on their own life. Emotional shared fate tracks affective responses to partners' outcomes. Specifically, the intensity of emotional shared fate corresponds to the appraisals people make about the extent to which others' positive and negative outcomes are likely to translate into a personal reward or loss. Examples include how good one feels when one's partner receives a job promotion, or how bad one feels when said partner's parents pass away [32].

Perceived and emotional shared fate inform people about the extent to which they are positively interdependent with specific relationship partners, which in turn predicts how much to value such partners and how much to invest in their welfare [32, 35]. Although the shared fate scale was developed as a six-item measure (with three items per subscale), here we employed

only one item per subscale to reduce participant fatigue and maximize the allocation of time and resources for other survey items. We selected the item "when [target] succeeds, I feel good" as the index of emotional shared fate because at Time 1 this item showed the strongest correlation with need-based helping attitude towards neighbors ($r$(303) = 0.34, $p < 0.001$), compared to other items ($r$'s = 0.07 to 0.28). Similarly, we selected the item "[target] and I rise and fall together" as the index of perceived shared fate because at Time 1 this item showed the strongest correlation with willingness to help a person from a different country ($r$(302) = 0.23, $p < 0.001$), compared to the other items ($r$'s = 0.01 to 0.22).

To determine the local prevalence of COVID-19, we matched participants' location information to the COVID-19 Data Repository maintained by the Center for Systems Science and Engineering (CSSE) at Johns Hopkins University [9]. We identified the latitude and longitude of each participants' postal code and matched their location information to the corresponding location in the COVID-19 Data repository to determine cumulative incidence rates of COVID-19 (i.e., confirmed cases per 100,000 individuals) of 520 participants. We were not able to match the prevalence of COVID-19 for participants who did not provide postal codes or who provided unidentifiable postal codes. Participants then reported their perceived risk of COVID-19 infection (*How likely do you think it is that you will contract/become infected with COVID-19?*; 1 = *not at all*, 7 = *extremely*). This measure positively correlates with objective markers of COVID-19, such as local prevalence of COVID-19 and local number of deaths caused by COVID-19 [88].

## Openness and transparency

This study was not preregistered. We report all data exclusions. Data, analysis code, and measures collected across the entire longitudinal survey not used in the present study are available at https://osf.io/gtsbr/?view_only=817bb93a937c4dabae23ad944d4b4da7.

## Results

### Analyses

We ran generalized mixed-effects linear models with a cumulative probit link function and maximum likelihood with Laplace approximation for the estimation method. Across models, we applied an unrestricted covariance structure, including a fixed and random intercept for participant ID, as well as a random effect for the slope of time. We included random effects for the slopes of COVID-19 prevalence, perceived risk of infection, perceived shared fate, and emotional shared fate whenever possible. We transformed COVID-19 prevalence scores using the base-10 logarithm to reduce skewness. We then computed cluster-mean scores (i.e., a person's average score on a given covariate across all time points) to obtain the between-person effects, and mean person-centered scores (i.e., a covariate's value at a given time point centered to each person's cluster-mean for that same covariate) to obtain the within-person effects. All covariates were standardized after centering. This approach allowed us to identify marginal changes over time and assess whether trait/chronic-level (i.e., between-person) and situational/person-centered (i.e., within-person) effects of predictors influenced the rate of change over time, while appropriately treating the dependent variables as ordinal measures. The time between successive data collection points was treated as one unit of time.

Because previous studies report both increases and decreases in cooperation and perceived interdependence following a crisis, we assessed linear and curvilinear changes over time in cooperation, shared fate, and perceived risk of infection. However, we only found a curvilinear effect of time for perceived infection risk, indicating that linear effects of time were a better fit to the data across all other measures (see Supplemental Information S2.1-S2.3). We then tested

**Table 2. Correlations among COVID-19 prevalence, perceived infection risk, shared fate, and cooperation.**

| Neighbors | COVID-19 | Infection risk | Emotional SF | Perceived SF | Will to help | Need-based help |
|---|---|---|---|---|---|---|
| COVID-19 prevalence | - | -.01 [-.03, .01] | .01 [-.01, .03] | .17 [.15, .19]*** | -.09 [-.11, -.07]*** | -.17 [-.19, -.15]*** |
| Perceived infection | .09 [.02, .15]** | - | .03 [.01, .05]* | .03 [.01, .06]** | .02 [.001, .04]* | .02 [.003, .05]* |
| Emotional shared fate | .04 [-.02, .10] | .04 [-.02, .10] | - | .42 [.40, .43]*** | .11 [.08, .13]*** | .17 [.15, .20]*** |
| Perceived shared fate | .07 [.004, .13]* | .08 [.01, .14]* | .70 [.67, .73]*** | - | .12 [.10, .15]*** | .08 [.06, .11]*** |
| Willingness to help | -.001 [-.06, .06] | -.02 [-.08, .04] | .33 [.27, .38]*** | .29 [.24, .35]*** | - | .26 [.24, .28]*** |
| Need-based helping | .05 [-.01, .12] | .05 [-.01, .12] | .49 [.45, .54]*** | .29 [.24, .35]*** | .39 [.34, .44]*** | - |
| **Humanity** | **COVID-19** | **Infection risk** | **Emotional SF** | **Perceived SF** | **Will to help** | **Need-based help** |
| COVID-19 prevalence | - | -.01 [-.03, .01] | -.14 [-.16, -.12]*** | .04 [.02, .07]*** | -.001 [-.02, .02] | -.17 [-.19, -.15]*** |
| Perceived infection | .09 [.02, .15]** | - | .05 [.02, .07]*** | .04 [.02, .07]*** | .02 [-.004, .04] | .003 [.003, .05]* |
| Emotional shared fate | .00002 [-.06, .06] | .05 [-.01, .11] | - | .35 [.33, .36]*** | .04 [.02, .06]*** | .18 [.16, .20]*** |
| Perceived shared fate | .01 [-.05, .07] | .10 [.04, .17]*** | .68 [.65, .72]*** | - | .07 [.05, .09]*** | .08 [.06, .11]*** |
| Willingness to help | .03 [-.03, .10] | .04 [-.02, .10] | .20 [.14, .26]*** | .19 [.13, .25]*** | - | 0.12 [.10, .14]*** |
| Need-based helping | .09 [.03, .16]** | .10 [.04, .16]** | .45 [.40, .50]*** | .27 [.21, .33]*** | .41 [.36, .46]*** | - |

*Note*. The upper diagonals show the within-person (i.e., repeated-measures) correlation. The lower diagonals show the between-person (i.e., scores averaged across time points) correlations. Numbers in brackets show 95% CIs.

*** = $p < 0.001$

** = $p < 0.01$

* = $p < 0.05$.

whether COVID-19 prevalence, perceived risk of infection, and shared fate influenced cooperation over time (S2.2 in S1 File), and whether COVID-19 prevalence and perceived risk of infection influenced perceived and emotional shared fate over time (S2.3 in S1 File). Last, we tested whether people reported higher cooperation and perceived interdependence with neighbors than with humanity (S2.4 in S1 File). We ran analyses with the GLIMMIX procedure for SAS V 9.4. Table 2 shows correlations among COVID-19 prevalence, perceived risk of infection, emotional shared fate, perceived shared fate, and inclinations to cooperate. We ran the repeated-measures correlations with the *Rmcorr* package [89] for R Studio (V. 2022.07.1).

## How did cooperation change during the pandemic?

**Inclinations to cooperate with neighbors decreased by a small margin over time.**   A one-unit increase in time was associated with a decrease in willingness to let a neighbor move into your home for a week ($b$ = -0.05, SE = 0.004, CI$_{95\%}$ [-0.06, -0.04]), leading to a decrease of $b$ = -0.65 (SE = 0.06, CI$_{95\%}$ [-0.77, -0.54]) by August 22, 2020 (S2.2.1 in S1 File; Fig 2A). A one-unit increase in time was associated with a decrease in agreeing with the statement that helping a neighbor in need is the right thing to do ($b$ = -0.04, SE = 0.005, CI$_{95\%}$ [-0.05, -0.03]), leading to a decrease of $b$ = -0.52 (SE = 0.06, CI$_{95\%}$ [-0.65, -0.39]) by August 22, 2020 (S2.2.2 in S1 File; Fig 2B).

**Inclinations to cooperate with humanity decreased by a small margin over time.**   A one-unit increase in time was associated with a decrease in willingness to let a person who is not a citizen of your own country move into your home for a week ($b$ = -0.02, SE = 0.005, CI$_{95\%}$ [-0.03, -0.01]), leading to a decrease of $b$ = -0.25 (SE = 0.07, CI$_{95\%}$ [-0.39, -0.11]) by August 22, 2020 (S2.2.3 in S1 File; Fig 2C). A one-unit increase in time was associated with a decrease in agreeing with the statement that helping a person who is not a citizen of your own country when they are in need is the right thing to do ($b$ = -0.02, SE = 0.004, CI$_{95\%}$ [-0.03, -0.02]), leading to a decrease of $b$ = -0.34 (SE = 0.06, CI$_{95\%}$ [-0.46, -0.22]) by August 22, 2020

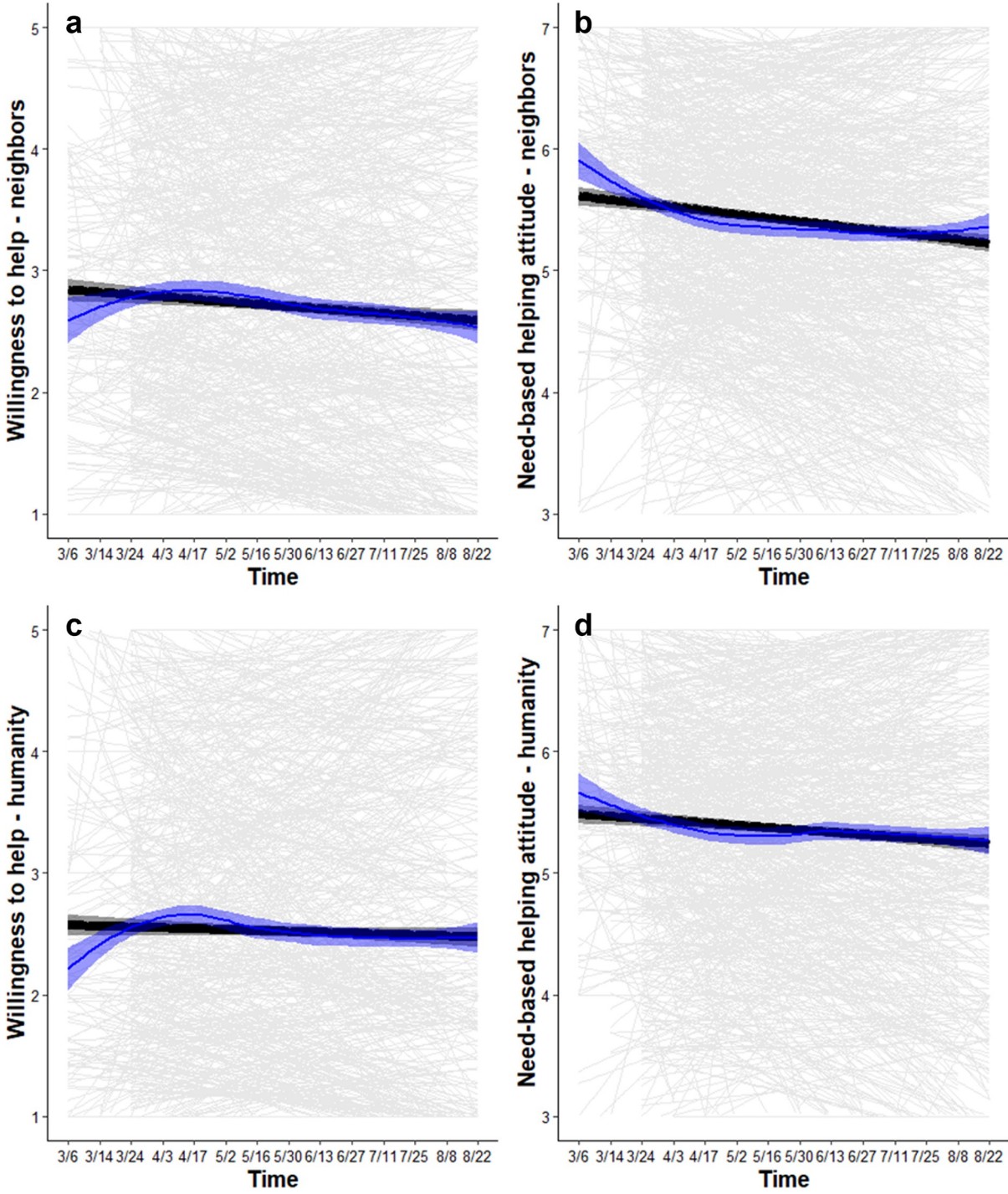

**Fig 2. Inclinations to cooperate from March-August 2020.** Black lines show the marginal effect of time, blue lines show the Loess curve (i.e., non-parametric line of best fit), and gray lines show the person-specific effect of time. Shaded bands show 95% CIs. Willingness to help (1 = *not at all willing*, 7 = *very willing*) neighbors (a) and humanity (c), as well as need-based helping attitude (1 = *do not agree at all*, 7 = *strongly agree*) toward neighbors (b) and humanity (d) decreased by a small margin over time.

(S2.2.4 in S1 File; Fig 2D). These results indicate that, excluding all covariates, inclinations to cooperate with neighbors and with humanity decreased by a small margin over time.

## How did perceived interdependence change during the pandemic?

**Shared fate with neighbors increased by a small margin over time.** A one-unit increase in time was associated with a small increase in emotional shared fate with neighbors (i.e., *When my neighborhood succeeds, I feel good*) ($b = 0.01$, SE = 0.004, CI$_{95\%}$ [0.003, 0.02]), leading to a small increase of $b = 0.14$ (SE = 0.05, CI$_{95\%}$ [0.04, 0.25]) by August 22, 2020 (S2.3.1 in S1 File; Fig 3A). A one-unit increase in time was associated with an increase in perceived shared fate with neighbors (i.e., *My neighborhood and I rise and fall together*) ($b = 0.05$, SE = 0.005, CI$_{95\%}$ [0.04, 0.06]), leading to an increase of $b = 0.66$ (SE = 0.06, CI$_{95\%}$ [0.53, 0.79]) by August 22, 2020 (S2.3.2 in S1 File; Fig 3B).

**Shared fate with humanity both increased and decreased by a small margin over time.** A one-unit increase in time was associated with a decrease in emotional shared fate with humanity (i.e., *When all of humanity succeeds, I feel good*) ($b = -0.04$, SE = 0.004, CI$_{95\%}$ [-0.04, -0.03]), leading to a decrease of $b = -0.48$ (SE = 0.05, CI$_{95\%}$ [-0.59, -0.37]) by August 22, 2020 (S2.3.3 in S1 File; Fig 3C). A one-unit increase in time was associated with a small increase in perceived shared fate with humanity (i.e., *All of humanity and I rise and fall together*) ($b = 0.01$, SE = 0.004, CI$_{95\%}$ [0.001, 0.02]), leading to a small increase of $b = 0.12$ (SE = 0.05, CI$_{95\%}$ [0.02, 0.23]) by August 22, 2020 (S2.3.4 in S1 File; Fig 3D). These results indicate that, excluding all covariates, shared fate with neighbors and humanity increased by a small margin over time, except for emotional shared fate with humanity, which began high at baseline and declined by a small margin over time.

## Did infection risk and perceived interdependence shift inclinations to cooperate over time?

In this section we report effects of COVID-19 prevalence, perceived risk of infection, and shared fate on cooperation at baseline (i.e., March 6, 2020) and over time. Within-person (i.e., Level-1) effects would indicate whether deviations in infection risk or shared fate, relative to individual's overall infection risk or shared fate, influenced cooperation at a given time point (i.e., observations shown in gray lines in Fig 2), whereas between-person (i.e., Level-2) effects at baseline would indicate whether infection risk or shared fate influenced the marginal intercept at baseline (i.e., intercept of black lines shown in Fig 2). Interactions with time at the within-person level would indicate whether covariates changed the person-specific effect of time (i.e., slope of gray lines shown in Fig 2). If we found that pattern of results, it would suggest that situation/occasion-specific perturbations in a covariate led to changes in cooperation over time. Interactions with time at the between-person level would indicate whether covariates changed the marginal effect of time (i.e., slope of black lines shown in Fig 2). If we found that pattern of results, it would instead suggest that pre-existing individual differences, or stable/chronic differences in settings across people, led to changes in cooperation over time.

**Perceived risk of COVID-19 infection shifted with COVID-19 prevalence.** A one-unit increase in time was associated with a small decrease in perceived risk of COVID-19 infection ($b = -0.01$, SE = 0.004, CI$_{95\%}$ [-0.02, -0.001]), leading to a decrease of $b = -0.11$ (SE = 0.05, CI$_{95\%}$ [-0.22, -0.01]) by August 22, 2020 (S2.1 in S1 File). Moreover, a time × time interaction ($b = 0.01$, SE = 0.003, CI$_{95\%}$ [0.006, 0.02]) indicated that, overall, perceived infection risk decreased from March to May but then increased from May to August 2020.

At the within-person level (Level-1), COVID-19 prevalence was associated with higher perceived infection risk, indicating that people perceived higher infection risk in time points in

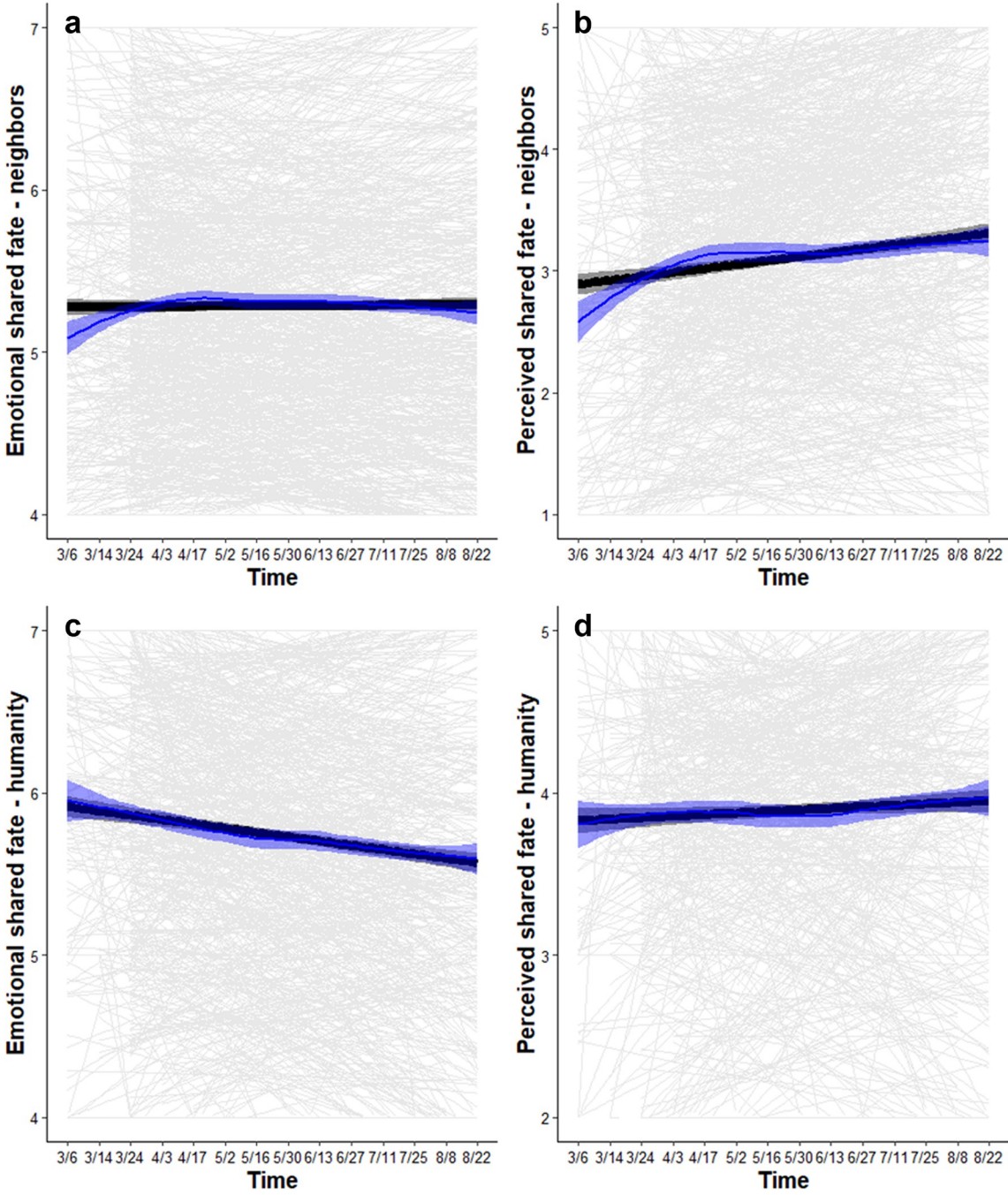

**Fig 3. Perceived interdependence from March-August 2020.** Black lines show the marginal effect of time, blue lines show the Loess curve (i.e., non-parametric line of best fit), and gray lines show the person-specific effect of time. Shaded bands show 95% CIs. Emotional (a) and perceived (b) shared fate with neighbors (1 = *do not agree at all*, 7 = *strongly agree*), and perceived shared fate with humanity (d) increased by a small margin over time. Emotional shared fate with humanity (c) decreased by a small margin over time.

which they experienced higher COVID-19 prevalence relative to their own location's overall COVID-19 prevalence ($b$ = 0.37, SE = 0.05, CI$_{95\%}$ [0.27, 0.48]). A time × COVID-19 prevalence interaction ($b$ = -0.16, SE = 0.02, CI$_{95\%}$ [-0.21, -0.11]) and a time × time × COVID-19 prevalence interaction ($b$ = 0.006, SE = 0.002, CI$_{95\%}$ [0.001, 0.01]) indicate that there was only a

quadratic effect of time when people experienced higher (+1SD) COVID-19 prevalence ($b$ = 0.02, SE = 0.001, CI$_{95\%}$ [0.01, 0.02]) but not when people experienced lower (-1SD) COVID-19 prevalence ($b$ = 0.006, SE = 0.005, CI$_{95\%}$ [-0.004, 0.01]). These results indicate that, when people experienced higher COVID-19 prevalence, relative to their own location's overall COVID-19 prevalence, people reported a sharper decrease in their perceived infection risk from March to June 2020 followed by an increase from June to August 2020.

At the between-person level (Level-2), COVID-19 prevalence was not associated with perceived infection risk on March 6, 2020 ($b$ = 0.03, SE = 0.06, CI$_{95\%}$ [-0.10, 0.16]). A time × COVID-19 prevalence interaction ($b$ = 0.05, SE = 0.02, CI$_{95\%}$ [0.02, 0.09]) and a time × time × COVID-19 prevalence interaction ($b$ = -0.003, SE = 0.001, CI$_{95\%}$ [-0.005, -0.001]) indicate that the quadratic effect of time was stronger for people who lived in an area with low (-1SD) COVID-19 prevalence ($b$ = 0.015, SE = 0.003, CI$_{95\%}$ [0.01, 0.02]) than for people who lived in an area with high (+1SD) COVID-19 prevalence ($b$ = 0.009, SE = 0.003, CI$_{95\%}$ [0.003, 0.01]). These results indicate that, compared to people who lived in areas with high COVID-19 prevalence, people who lived in areas with low COVID-19 prevalence reported a sharper decrease in their perceived infection risk from March to May followed by a sharper increase from May to August 2020. This model (Table S2; S2.1 in S1 File) accounted for 67.60% of the between-person (i.e., Level-2) variance, and 6.22% of the within-person (i.e., Level-1) variance (ICC = 0.35).

**Willingness to help neighbors shifted with COVID-19 prevalence, perceived infection risk, and shared fate.** *COVID-19 prevalence*. At the within-person level, COVID-19 prevalence was not associated with willingness to help neighbors ($b$ = 0.01, SE = 0.03, CI$_{95\%}$ [-0.05, 0.08]), indicating that deviations in COVID-19 prevalence, relative to one's location overall COVID-19 prevalence, were not associated with willingness to help neighbors. However, a time × COVID-19 prevalence interaction ($b$ = -0.05, SE = 0.01, CI$_{95\%}$ [-0.07, -0.02]) shows that willingness to help neighbors decreased over time when people, relative to their own location's overall COVID-19 prevalence, experienced higher (+1SD) COVID-19 prevalence ($b$ = -0.10, SE = 0.01, CI$_{95\%}$ [-0.13, -0.07]) but not when experiencing lower (-1SD) COVID-19 prevalence ($b$ = -0.01, SE = 0.02, CI$_{95\%}$ [-0.06, 0.04]) (Fig 4A).

At the between-person level, COVID-19 prevalence was not associated with willingness to help neighbors on March 6, 2020 ($b$ = 0.04, SE = 0.04, CI$_{95\%}$ [-0.04, 0.12]), indicating that COVID-19 prevalence was not associated with the marginal intercept. A time × COVID-19 prevalence interaction ($b$ = -0.02, SE = 0.006, CI$_{95\%}$ [-0.03, -0.01]) shows that willingness to help neighbors decreased over time for people who lived in areas with high (+1SD) COVID-19 prevalence ($b$ = -0.08, SE = 0.02, CI$_{95\%}$ [-0.11, -0.04]), but not for people who lived in areas with low (-1SD) COVID-19 prevalence ($b$ = -0.03, SE = 0.02, CI$_{95\%}$ [-0.07, 0.0002]) (Fig 4B).

*Perceived risk of infection*. At the within-person level, perceived infection risk was not associated with willingness to help neighbors ($b$ = 0.003, SE = 0.01, CI$_{95\%}$ [-0.02, 0.03]), indicating that deviations in perceived infection risk, relative to one's own overall perceived infection risk, were not associated with willingness to help neighbors. At the between-person level, perceived infection risk was not associated with willingness to help on March 6, 2020 ($b$ = 0.04, SE = 0.04, CI$_{95\%}$ [-0.04, 0.12]), indicating that perceived infection risk was not associated with the marginal intercept. However, a time × perceived infection risk interaction ($b$ = -0.02, SE = 0.006, CI$_{95\%}$ [-0.03, -0.01]) shows that willingness to help neighbors decreased at a faster rate over time for people with high (+1SD) trait-perceived infection risk ($b$ = -0.08, SE = 0.02, CI$_{95\%}$ [-0.11, -0.04]) than for people with low (-1SD) trait-perceived infection risk ($b$ = -0.03, SE = 0.01, CI$_{95\%}$ [-0.06, -0.002]) (Fig 4C).

*Shared fate*. At the within-person level, perceived ($b$ = 0.07, SE = 0.01, CI$_{95\%}$ [0.04, 0.10]) and emotional shared fate ($b$ = 0.04, SE = 0.01, CI$_{95\%}$ [0.01, 0.07]) were associated with higher

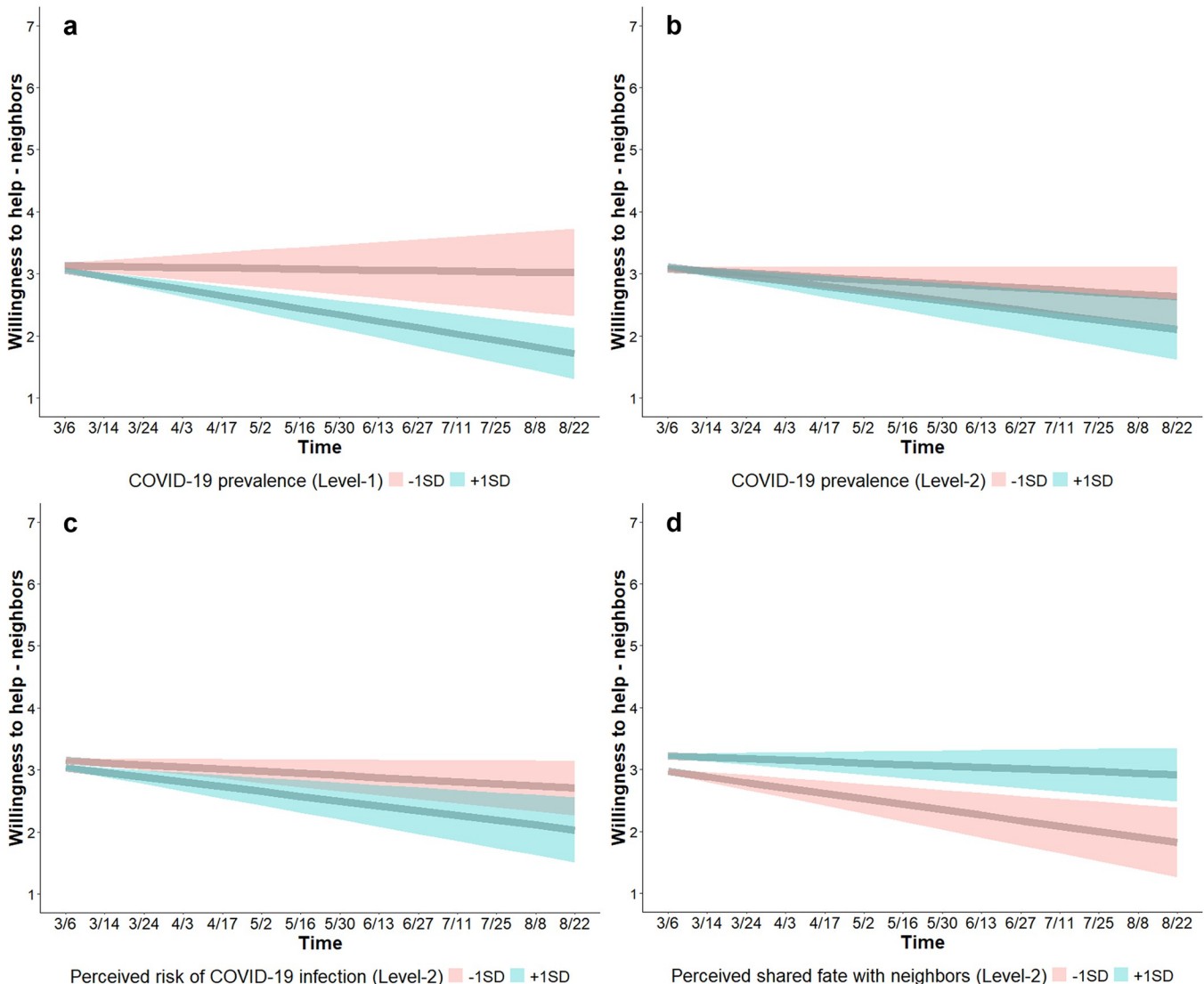

**Fig 4. Changes in willingness to help neighbors.** Willingness to help neighbors (*Someone from your neighborhood is having their residence fixed, so it isn't livable. How willing would you be to let them move into your residence for a week*?; 1 = *not at all willing*, 7 = *very willing*) decreased over time when people experienced higher (blue; a), but not when people experienced lower (red; a), COVID-19 prevalence. Willingness to help decreased over time for people who lived in areas with high (blue; b), but not low (red; b), COVID-19 prevalence. Willingness to help decreased more over time for people with high (blue; c), than for people with low (red; c), trait perceived infection risk. Willingness to help decreased over time for people with low (red; d), but not for people with high (blue; d), perceived shared fate with neighbors. Shaded bands show 95% CIs.

willingness to help, indicating that people reported higher willingness to help neighbors in time points in which people experienced higher perceived and emotional shared fate with neighbors relative to their own overall perceived and emotional shared fate with neighbors. At the between-person level, emotional shared fate was associated with higher willingness to help at baseline ($b = 0.28$, SE = 0.05, $CI_{95\%}$ [0.17, 0.39]), indicating that emotional shared fate was associated with a higher marginal intercept. Perceived shared fate was not associated with willingness to help at baseline ($b = 0.09$, SE = 0.06, $CI_{95\%}$ [-0.02, 0.20]), but a time × perceived shared fate interaction ($b = 0.03$, SE = 0.01, $CI_{95\%}$ [0.01, 0.05]) indicates that willingness to help neighbors decreased over time for people with low (-1SD) perceived shared fate with

neighbors ($b$ = -0.09, SE = 0.02, CI$_{95\%}$ [-0.13, -0.05]) but not for people with high (+1SD) perceived shared fate with neighbors ($b$ = -0.02, SE = 0.01, CI$_{95\%}$ [-0.05, 0.01]) (Fig 4D). This model (Table S3; S2.2.1 in S1 File) accounted for 78.55% of the between-person variance, but none of the within-person variance in willingness to help neighbors (ICC = 0.49).

**Need-based helping attitude toward neighbors shifted with perceived infection risk and shared fate.** *COVID-19 prevalence.* At the within-person level, COVID-19 prevalence was associated with lower need-based helping attitude toward neighbors ($b$ = -0.28, SE = 0.03, CI$_{95\%}$ [-0.35, -0.21]), indicating that people reported lower need-based helping attitude toward neighbors in time points in which they experienced higher COVID-19 prevalence relative to their own location's overall COVID-19 prevalence. At the between-person level, COVID-19 prevalence was associated with higher need-based helping attitude toward neighbors at baseline ($b$ = 0.17, SE = 0.04, CI$_{95\%}$ [0.09, 0.25]). COVID-19 prevalence was not associated with changes in need-based helping attitude over time (Table S4 in S1 File). These results indicate that people who experienced higher overall COVID-19 prevalence reported lower need-based helping attitude at baseline, and that people reported lower need-based helping attitude toward neighbors in months in which they experienced higher COVID-19, but these effects did not change over time.

*Perceived risk of infection.* At the within-person level, perceived infection risk was not associated with need-based helping attitude (Table S4 in S1 File). At the between-person level, perceived infection risk was associated with higher need-based helping attitude at baseline ($b$ = 0.11, SE = 0.02, CI$_{95\%}$ [0.02, 0.20]). A time × perceived infection risk interaction ($b$ = -0.01, SE = 0.002, CI$_{95\%}$ [-0.02, -0.006]) shows that need-based helping attitude decreased over time for people with high (+1SD) trait-perceived infection risk ($b$ = -0.02, SE = 0.01, CI$_{95\%}$ [-0.03, -0.003]), but not for people with low (-1SD) trait-perceived infection risk ($b$ = 0.01, SE = 0.01, CI$_{95\%}$ [-0.003, 0.03]) (Fig 5A).

*Shared fate.* At the within-person level, emotional shared fate ($b$ = 0.14, SE = 0.01, CI$_{95\%}$ [0.11, 0.17]) but not perceived shared fate ($b$ = -0.04, SE = 0.03, CI$_{95\%}$ [-0.10, 0.03]) was

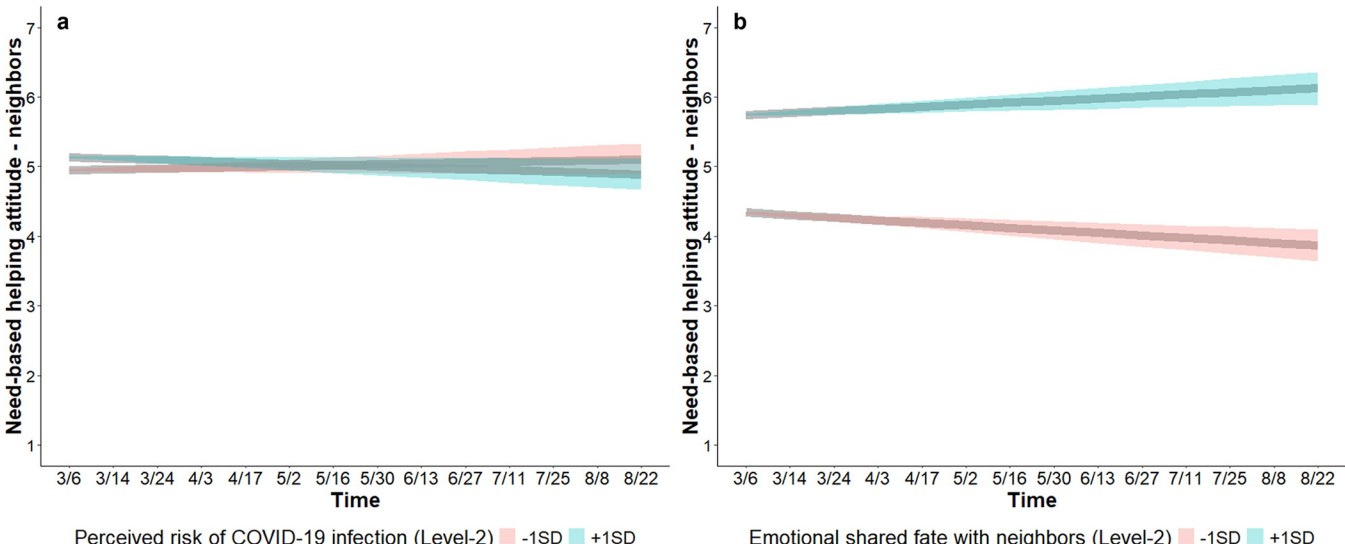

**Fig 5. Changes in need-based helping attitude toward neighbors.** Need-based helping attitude (*Helping someone from my neighborhood when they are in need is the right thing to do*; 1 = strongly disagree, 7 = strongly agree) decreased by a small margin over time for people with high trait perceived infection risk (blue; a), but not for people with low trait perceived infection risk (red; a). Need-based helping attitude decreased over time for people with low emotional shared fate with neighbors (red; b), but increased over time for people with high emotional shared fate with neighbors (blue; b). Shaded bands show 95% CIs.

associated with higher need-based helping attitude, indicating that people reported higher need-based helping attitude in time points in which they experienced higher emotional shared fate with neighbors relative to their own overall emotional shared fate with neighbors. Emotional and perceived shared fate were not associated with changes over time (Table S4 in S1 File). At the between-person level, perceived shared fate was associated with lower need-based helping attitude at baseline ($b$ = -0.17, SE = 0.06, $CI_{95\%}$ [-0.29, -0.06]), while emotional shared fate was associated with higher ($b$ = 0.67, SE = 0.07, $CI_{95\%}$ [0.53, 0.80]) need-based helping attitude at baseline. A time × emotional shared fate interaction ($b$ = 0.03, SE = 0.005, $CI_{95\%}$ [0.02, 0.04]) shows that need-based helping attitude increased over time for people with high (+1SD) emotional shared fate ($b$ = 0.03, SE = 0.01, $CI_{95\%}$ [0.01, 0.04]) but decreased over time for people with low (-1SD) emotional shared fate with neighbors ($b$ = -0.03, SE = 0.01, $CI_{95\%}$ [-0.05, -0.02]) (Fig 5B). This model (Table S4; S2.2.2 in S1 File) accounted for 49.58% of the between-person (i.e., Level-2) variance, and 13.63% of the within-person (i.e., Level-1) variance (ICC = 0.38).

**Willingness to help a person from a different country shifted with COVID-19 prevalence and shared fate.** *COVID-19 prevalence.* At the within-person level, COVID-19 prevalence was associated with higher willingness to help a person from a different country ($b$ = 0.11, SE = 0.04, $CI_{95\%}$ [0.03, 0.19]), indicating that people reported higher willingness to help a person from a different country in time points in which they experienced higher COVID-19 prevalence. A time × COVID-19 prevalence interaction ($b$ = -0.04, SE = 0.01, $CI_{95\%}$ [-0.07, -0.01]) shows that willingness to help a person from a different country decreased over time when people, relative to their own location's overall COVID-19 prevalence, experienced higher (+1SD) COVID-19 prevalence ($b$ = -0.11, SE = 0.03, $CI_{95\%}$ [-0.18, -0.04]) but not when people experienced lower (-1SD) COVID-19 prevalence ($b$ = -0.03, SE = 0.03, $CI_{95\%}$ [-0.09, 0.02]) (Fig 6A).

At the between-person level, COVID-19 prevalence was not associated with willingness to help ($b$ = 0.06, SE = 0.04, $CI_{95\%}$ [-0.02, 0.15]). A time × COVID-19 prevalence interaction ($b$ = -0.02, SE = 0.01, $CI_{95\%}$ [-0.03, -0.0003]) shows that willingness to help decreased at a faster rate over time for people who lived in an area with high (+1SD) COVID-19 prevalence ($b$ = -0.09, SE = 0.03, $CI_{95\%}$ [-0.15, -0.02]) than for people who lived in an area with low (-1SD) COVID-19 prevalence ($b$ = -0.05, SE = 0.02, $CI_{95\%}$ [-0.10, -0.005]) (Fig 6B).

*Perceived infection risk.* Perceived infection risk was not associated with willingness to help a person from a different country at baseline or over time (Table S5 in S1 File).

*Shared fate.* At the within-person level, perceived shared fate was associated with higher willingness to help ($b$ = 0.05, SE = 0.01, $CI_{95\%}$ [0.02, 0.07]), indicating that people reported higher willingness to help in time points in which they perceived higher shared fate with humanity. Emotional shared fate was not associated with willingness to help, and neither emotional nor perceived shared fate were associated with willingness to help over time (Table S5 in S1 File). At the between-person level, emotional shared fate ($b$ = 0.15, SE = 0.06, $CI_{95\%}$ [0.03, 0.26]), but not perceived shared fate ($b$ = 0.07, SE = 0.06, $CI_{95\%}$ [-0.05, 0.18]), was associated with higher willingness to help at baseline. A time × perceived shared fate interaction ($b$ = 0.03, SE = 0.01, $CI_{95\%}$ [0.01, 0.05]) shows that willingness to help a person from a different country decreased over time for people with low (-1SD) perceived shared fate with humanity ($b$ = -0.10, SE = 0.04, $CI_{95\%}$ [-0.17, -0.03]) but not for people with high (+1SD) perceived shared fate with humanity ($b$ = -0.04, SE = 0.02, $CI_{95\%}$ [-0.08, 0.0004]) (Fig 6C). This model (Table S5; S2.2.3 in S1 File) accounted for 78.96% of the between-person (i.e., Level-2) variance, and 0.20% of the within-person (i.e., Level-1) variance (ICC = 0.51).

**Need-based helping attitude towards a person from a different country shifted with COVID-19 prevalence and shared fate.** *COVID-19 prevalence.* At the within-person level,

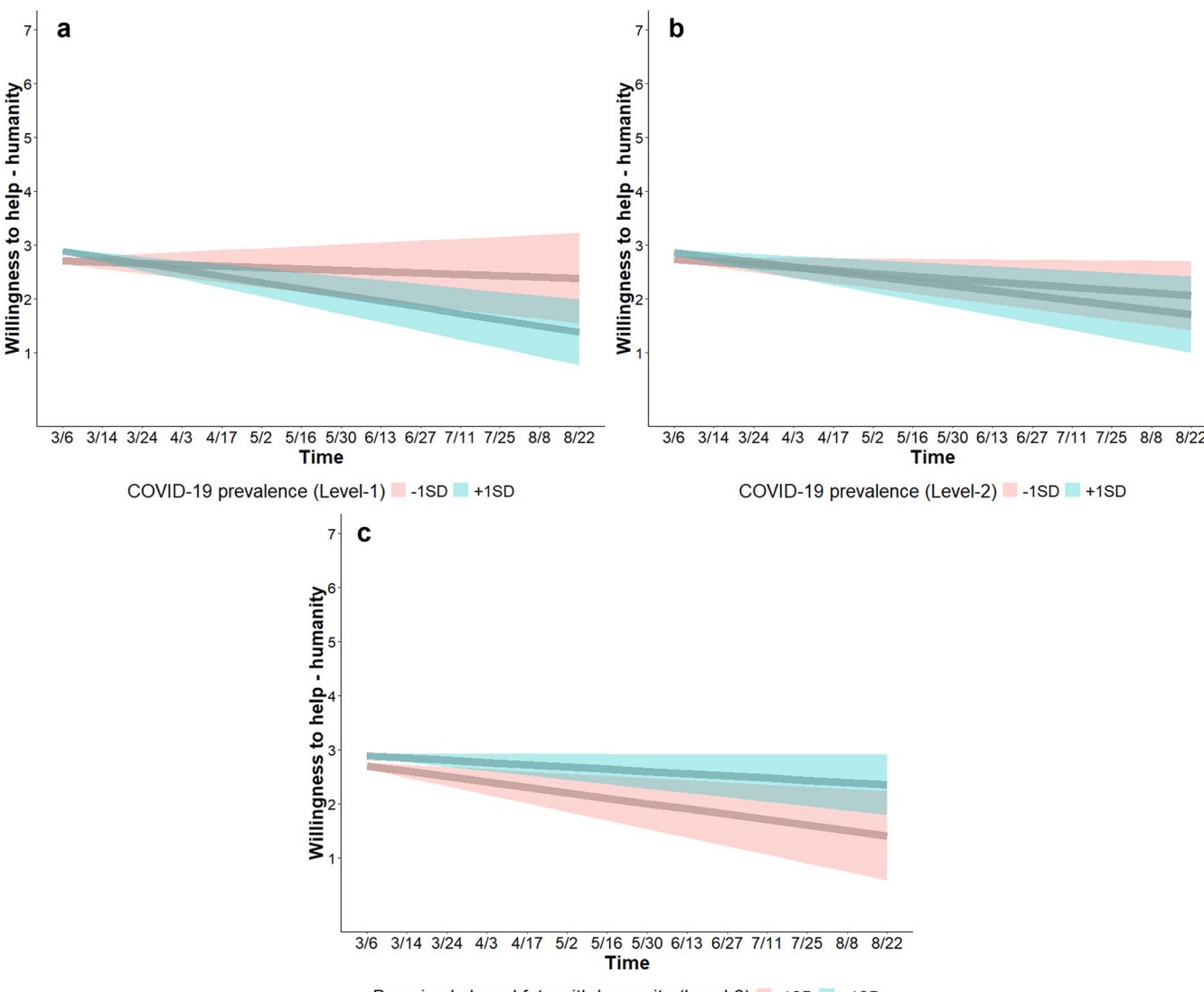

**Fig 6. Changes in willingness to help a person from a different country.** Willingness to help (*A person who is not a citizen of your own country is having their residence fixed, so it isn't livable. How willing would you be to let them move into your residence for a week?*; 1 = *not at all willing*, 7 = *very willing*) decreased over time when people experienced higher COVID-19 prevalence (blue; a), but not when people experienced lower COVID-19 prevalence (red; a). Willingness to help decreased more over time for people who lived in areas with high COVID-19 prevalence (blue; b), than for people who lived in areas with low COVID-19 prevalence (red; b). Willingness to help decreased over time for people with low perceived shared fate with humanity (red; c) but for people with high perceived shared fate with humanity (blue; c). Shaded bands show 95% CIs.

COVID-19 prevalence was associated with lower need-based helping attitude ($b$ = -0.15, SE = 0.04, $CI_{95\%}$ [-0.22, -0.07]), indicating that people reported lower need-based helping attitude towards a person from a different country in time points in which they experienced higher COVID-19 prevalence. A time × COVID-19 prevalence interaction ($b$ = -0.03, SE = 0.01, $CI_{95\%}$ [-0.06, -0.01]) shows that need-based helping attitude towards a person from a different country increased over time when people, relative to their own location's overall COVID-19 prevalence, experienced lower (-1SD) COVID-19 prevalence ($b$ = 0.08, SE = 0.02, $CI_{95\%}$ [0.03, 0.13]) but not when people experienced higher (+1SD) COVID-19 prevalence ($b$ = 0.01, SE = 0.01, $CI_{95\%}$ [-0.01, 0.02]) (Fig 7A). At the between-person level, COVID-19

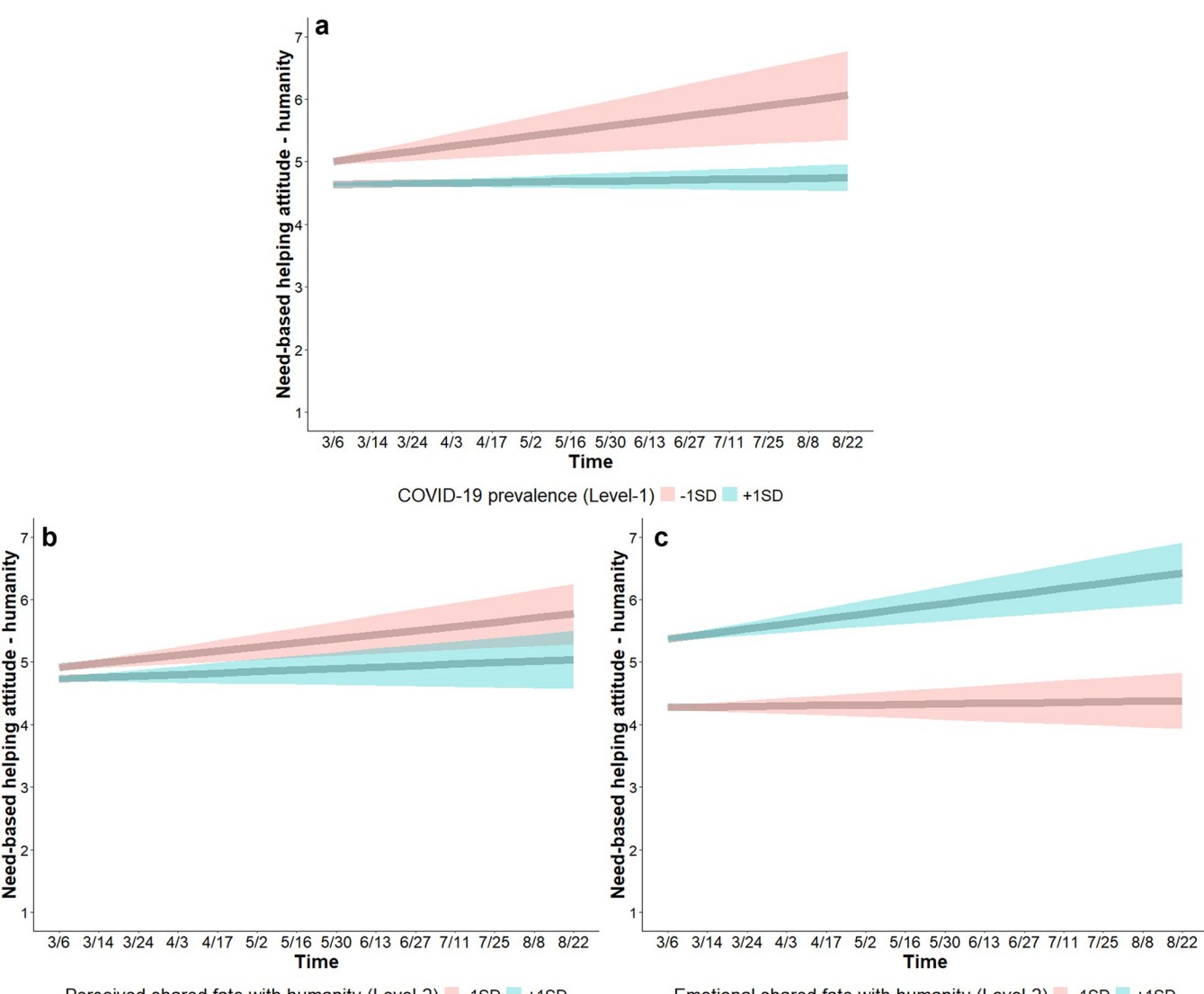

**Fig 7. Changes in need-based helping attitude towards a person from a different country.** Need-based helping attitude (*Helping a person who is not a citizen of your own country when they are in need is the right thing to do*; 1 = *strongly disagree*, 7 = *strongly agree*) increased over time when people experienced lower COVID-19 prevalence (red; a), but not when people experienced higher COVID-19 prevalence (blue; a). Need-based helping attitude increased over time for people with low perceived shared fate with humanity (red; b), but not for people with high perceived shared fate with humanity (blue; b). Need-based helping attitude increased over time for people with high emotional shared fate with humanity (blue; c), but not for people with low emotional shared fate with humanity (red; c). Shaded bands show 95% CIs.

prevalence was associated with higher need-based helping attitude at baseline ($b = 0.11$, SE = 0.03, CI$_{95\%}$ [0.04, 0.18]), but not with changes in need-based helping attitude over time (Table S6 in S1 File).

*Perceived risk of infection.* At the within-person level, perceived infection risk was not associated with need-based helping attitude towards a person from a different (Table S6 in S1 File). At the between-person level, perceived infection risk was associated with higher need-based helping attitude at baseline ($b = 0.07$, SE = 0.03, CI$_{95\%}$ [0.001, 0.14]) but not over time (Table S6 in S1 File).

*Shared fate.* At the within-person level, emotional shared fate was associated with higher need-based helping attitude ($b = 0.11$, SE = 0.01, CI$_{95\%}$ [0.07, 0.14]), indicating that people

reported higher need-based helping attitude in time points in which they experienced higher emotional shared fate with humanity. Perceived shared fate was not associated with need-based helping attitude at baseline, and neither emotional nor perceived shared fate were associated with need-based helping attitude over time (Table S6 in S1 File).

At the between-person level, perceived shared fate was not associated with need-based helping attitude at baseline ($b$ = -0.07, SE = 0.06, CI$_{95\%}$ [-0.19, 0.05]). In contrast to our expectation, a time × perceived shared fate interaction ($b$ = -0.02, SE = 0.01, CI$_{95\%}$ [-0.04, -0.002]) shows that need-based helping attitude towards a person from a different country increased over time for people with low (-1SD) perceived shared fate with humanity ($b$ = 0.06, SE = 0.02, CI$_{95\%}$ [0.03, 0.10]) but not for people with high (+1SD) perceived shared fate with humanity ($b$ = 0.02, SE = 0.01, CI$_{95\%}$ [-0.01, 0.05]) (Fig 7B).

Emotional shared fate was associated with higher need-based helping attitude at baseline ($b$ = 0.51, SE = 0.06, CI$_{95\%}$ [0.39, 0.62]). A time × emotional shared fate interaction ($b$ = 0.03, SE = 0.01, CI$_{95\%}$ [0.02, 0.05]) shows that need-based helping attitude towards a person from a different country increased over time for people with high (+1SD) emotional shared fate with humanity ($b$ = 0.08, SE = 0.02, CI$_{95\%}$ [0.04, 0.11]) but not for people with low (-1SD) emotional shared fate with humanity ($b$ = 0.01, SE = 0.01, CI$_{95\%}$ [-0.02, 0.04]) (Fig 7C). This model (Table S6; S2.2.4 in S1 File) accounted for 72.06% of the between-person (i.e., Level-2) variance, and 14.41% of the within-person (i.e., Level-1) variance (ICC = 0.38). See Table 3 for a summary of the changes in inclinations to cooperate.

## Did infection risk influence perceived interdependence over time?

**Emotional shared fate with neighbors increased over time when people experienced lower COVID-19 prevalence.** At the within-person level, COVID-19 prevalence was not associated with emotional shared fate with neighbors ($b$ = 0.06, SE = 0.04, CI$_{95\%}$ [-0.02, 0.15]). However, a time × COVID-19 prevalence interaction ($b$ = -0.03, SE = 0.01, CI$_{95\%}$ [-0.05, -0.01]) shows that emotional shared fate with neighbors increased over time when, relative to their own location's overall COVID-19 prevalence, people experienced lower (-1SD) COVID-19 prevalence ($b$ = 0.05, SE = 0.02, CI$_{95\%}$ [0.01, 0.10]) but not when people experienced higher (+1SD) COVID-19 prevalence ($b$ = -0.01, SE = 0.006, CI$_{95\%}$ [-0.02, -0.00004]) (Fig 8A). In contrast to predictions, perceived infection risk was associated with higher emotional shared fate with neighbors ($b$ = 0.03, SE = 0.01, CI$_{95\%}$ [0.003, 0.05]), indicating that people reported higher emotional shared fate with neighbors in time points in which they perceived higher infection risk, but perceived infection risk did not influence emotional shared fate over time (Table S7 in S1 File). At the between-person level, COVID-19 prevalence and perceived infection risk were not associated with emotional shared fate with neighbors at baseline or over time (Table S7 in S1 File). This model (Table S7; S2.3.1 in S1 File) accounted for 22.20% of the between-person (i.e., Level-2) variance, and 16.11% of the within-person (i.e., Level-1) variance (ICC = 0.44).

**Perceived shared fate with neighbors increased over time when people experienced lower COVID-19 prevalence.** At the within-person level, COVID-19 prevalence was associated with higher perceived shared fate with neighbors ($b$ = 0.37, SE = 0.03, CI$_{95\%}$ [0.30, 0.45]), indicating that people perceived higher shared fate with neighbors in time points in which they experienced higher COVID-19 prevalence. However, over time, a time × COVID-19 prevalence interaction ($b$ = -0.04, SE = 0.01, CI$_{95\%}$ [-0.06, -0.01]) shows that perceived shared fate with neighbors increased when people experienced lower (-1SD) COVID-19 prevalence ($b$ = 0.07, SE = 0.02, CI$_{95\%}$ [0.03, 0.11]) but not when people experienced higher (+1SD) COVID-19 prevalence ($b$ = -0.003, SE = 0.006, CI$_{95\%}$ [-0.01, 0.01]) (Fig 8B). Perceived infection risk was not associated with perceived shared fate (Table S8 in S1 File).

**Table 3. Summary of the changes in inclinations to cooperate from March to August 2020.**

| Cooperation items | T1 M (SD) | Effect of time — Change by T14 | Effect of COVID-19 prevalence — Baseline/time-varying | Effect of COVID-19 prevalence — Time × COVID | Effect of perceived risk of infection — Baseline/time-varying | Effect of perceived risk of infection — Time × perceived risk | Effect of shared fate — Baseline/time-varying | Effect of shared fate — Time × shared fate |
|---|---|---|---|---|---|---|---|---|
| Someone from your **neighborhood** is having their house fixed, so it isn't livable. How willing would you be to let them move into your house for a week? | 3.11 (1.78) | b = -.654 [-.771, -.537]*** | – | *Level-2* COVID (-1SD) b = -.034 [-.068, .0002] *Level-2* COVID (+1SD) b = -.077 [-.112, -.043]*** *Level-1* COVID (-1SD) b = -.008 [-.058, .042] *Level-1* COVID (+1SD) b = -.103 [-.132, -.074]*** | – | *Level-2* Perceived risk (-1SD) b = -.034 [-.065, -.002]* Perceived risk (+1SD) b = -.077 [-.115, -.040]*** | *Level-2* (baseline) Emotional SF b = .280 [.169, .390]*** *Level-1* (time-varying) Perceived SF b = .071 [.042, .099]*** Emotional SF b = .044 [.015, .073]** | *Level-2* Perceived SF (-1SD) b = -.088 [-.128, .048]*** Perceived SF (+1SD) b = -.023 [-.054, .007] |
| Helping someone from your **neighborhood** when they are in need is the right thing to do | 5.63 (1.40) | b = -.521 [-.647, -.395]*** | *Level-2* (baseline) b = .171 [.093, .249]*** *Level-1* (time-varying) b = -.281 [-.349, -.212]*** | – | *Level-2* (baseline) b = .111 [.017, .206]*** | *Level-2* Perceived risk (-1SD) b = .012 [-.003, .028] Perceived risk (+1SD) b = -.019 [-.035, -.003]* | *Level-2* (baseline) Perceived SF b = -.174 [-.289, -.059]** Emotional SF b = .668 [.532, .804]*** *Level-1* (time-varying) Emotional SF b = .139 [.111, .167]** | *Level-2* Emotional SF (-1SD) b = -.036 [-.052, -.020]*** Emotional SF (+1SD) b = .029 [.013, .046]*** |
| A person who is **not a citizen of your own country** is having their house fixed, so it isn't livable. How willing would you be to let them move into your house for a week? | 2.39 (1.61) | b = -.249, [-.388, -.111]*** | *Level-1* (time-varying) b = .116 [.036, .196]** | *Level-2* COVID (-1SD) b = -.055 [-.105, -.005]* *Level-2* COVID (+1SD) b = -.090 [-.156, -.024]** *Level-1* COVID (-1SD) b = -.031 [-.089, .026] *Level-1* COVID (+1SD) b = -.113 [-.182, -.045]** | – | – | *Level-2* (baseline) Emotional SF b = .149 [.036, .260]** *Level-1* (time-varying) Perceived SF b = .048 [.020, .075]** | *Level-2* Perceived SF (-1SD) b = -.102 [-.175, .028]** Perceived SF (+1SD) b = -.043 [-.087, .0004] |
| Helping a person who is **not a citizen of your own country** when they are in need is the right thing to do | 5.42 (1.38) | b = -.344 [-.462, -.227]*** | *Level-2* (baseline) b = .111 [.040, .183]** *Level-1* (time-varying) b = -.149 [-.225, -.072]*** | *Level-1* COVID (-1SD) b = .081 [.030, .132]** *Level-1* COVID (+1SD) b = .008 [-.007, .023] | *Level-2* (baseline) b = .074 [.002, .147]* | – | *Level-2* (baseline) Emotional SF b = .511 [.396, .626]*** *Level-1* (time-varying) Emotional SF b = .108 [.076, .140]*** | *Level-2* Perceived SF (-1SD) b = .065 [.031, .10]*** Perceived SF (+1SD) b = .023 [-.009, .056] Emotional SF (-1SD) b = .008 [-.023, .040] Emotional SF (+1SD) b = .081 [.046, .116]*** |

*Note.* Cooperation items were scored on 7-point scales (1 = *not at all willing/do not agree at all*, 7 = *very willing/strongly agree*). We tested whether prevalence of COVID-19, perceived infection risk, and perceived interdependence were associated with cooperation at baseline and with changes over time. SF = shared fate. Level-2 indicates between-person effects, and Level-1 indicates within-person effects derived from GLMMs. The numbers in brackets represent 95% CIs.

*** = $p < 0.001$

** = $p < 0.01$

* = $p < 0.05$.

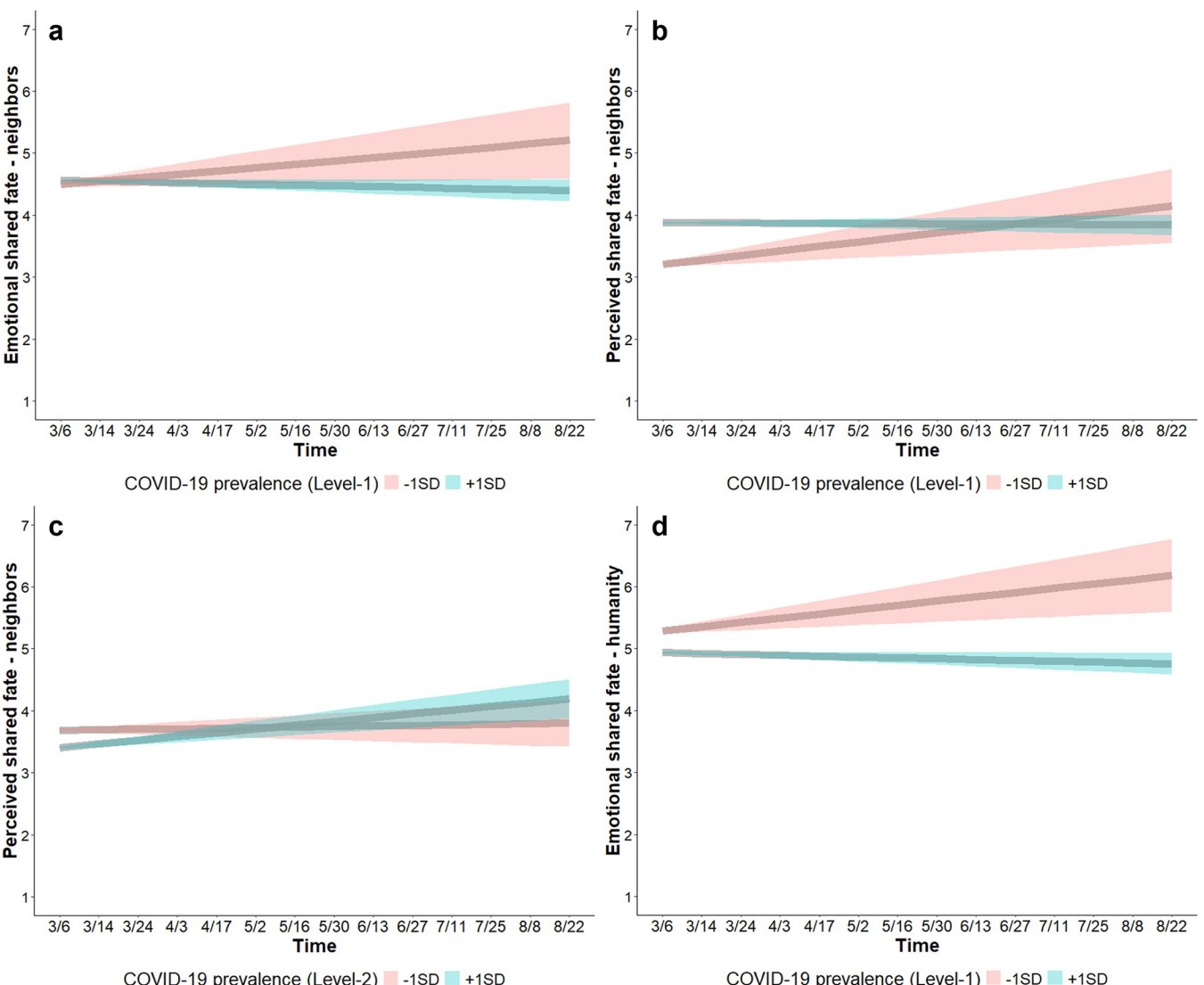

**Fig 8. Changes in perceived interdependence.** All items were scored on 7-point scales (1 = *do not agree at all*, 7 = *strongly agree*). Emotional shared fate (*When my neighborhood succeeds, I feel good*) and perceived shared fate with neighbors (*My neighborhood and I rise and fall together*) increased over time when people experienced lower COVID-19 prevalence (red; a-b), but not when people experienced higher COVID-19 prevalence (blue; a-b). Perceived shared fate with neighbors increased over time for people residing in high COVID-19 prevalence areas (blue; c), but not for people in low COVID-19 prevalence areas (red; c). Emotional shared fate with humanity (*When All of humanity succeeds, I feel good*) increased when people experienced lower COVID-19 prevalence (red; d), but decreased by a small margin over time when people experienced higher COVID-19 prevalence (blue; d). Shaded bands show 95% CIs.

At the between-person level, COVID-19 prevalence was associated with lower perceived shared fate at baseline ($b$ = -0.17, SE = 0.05, CI$_{95\%}$ [-0.27, -0.06]). Contrary to predictions, a time × COVID-19 prevalence interaction ($b$ = 0.02, SE = 0.005, CI$_{95\%}$ [0.01, 0.03]) shows that perceived shared fate with neighbors increased over time for people who lived in areas with high (+1SD) COVID-19 prevalence ($b$ = 0.06, SE = 0.01, CI$_{95\%}$ [0.04, 0.08]) but not for people who lived in areas with low (-1SD) COVID-19 prevalence ($b$ = 0.01, SE = 0.01, CI$_{95\%}$ [-0.02, 0.03]) (Fig 8C). Perceived infection risk was not associated with perceived shared fate at baseline or over time (Table S8 in S1 File). This model (Table S8; S2.3.2 in S1 File) accounted for 27.58% of the between-person (i.e., Level-2) variance, and 22.15% of the within-person (i.e., Level-1) variance (ICC = 0.40).

**Emotional shared fate with humanity increased over time when people experienced lower COVID-19 prevalence.** At the within-person level, COVID-19 prevalence was associated with lower emotional shared fate with humanity ($b$ = -0.13, SE = 0.03, CI$_{95\%}$ [-0.20, -0.06]), indicating that people reported lower emotional shared fate with humanity in time points in which they experienced higher COVID-19 prevalence. A time × COVID-19 prevalence interaction ($b$ = -0.04, SE = 0.01, CI$_{95\%}$ [-0.06, -0.02]) shows that emotional shared fate with humanity increased over time when, relative to their own location's overall COVID-19 prevalence, people experienced lower (-1SD) COVID-19 prevalence ($b$ = 0.07, SE = 0.02, CI$_{95\%}$ [0.03, 0.11]), but decreased by a small margin over time when people experienced higher (+1SD) COVID-19 prevalence ($b$ = -0.01, SE = 0.006, CI$_{95\%}$ [-0.03, -0.001]) (Fig 8D). Perceived infection risk was not associated with emotional shared fate with humanity (Table S9 in S1 File). At the between-person level COVID-19 prevalence and perceived infection risk were not associated with emotional shared fate with humanity at baseline or over time (Table S9 in S1 File). This model (Table S9; S2.3.3 in S1 File) accounted for 48.82% of the between-person (i.e., Level-2) variance, and 8.50% of the within-person (i.e., Level-1) variance (ICC = 0.42).

**Perceived shared fate with all of humanity was not associated with infection risk.** COVID-19 prevalence and perceived infection risk were not associated with perceived shared fate with humanity (Table S10; S2.3.4 in S1 File). The effect of time accounted for 26.69% of the between-person (i.e., Level-2) variance and 1.12% of the within-person (i.e., Level-1) variance (ICC = 0.41). See Table S11 in S1 File for a summary of the changes in perceived interdependence.

## People reported higher cooperation but lower interdependence with their neighbors than with humanity

We ran paired samples $t$-tests (two-tailed) at baseline, middle, and end of the survey (Tables S12 and S13; S2.4 in S1 File). As expected, people reported greater willingness to help (Fig 9A) and higher need-based helping attitude (Fig 9B) towards neighbors than a person from a different country. In contrast to our expectations, however, people reported higher emotional shared fate (Fig 9C) and perceived shared fate (Fig 9D) with humanity than with neighbors.

## Discussion

Did the COVID-19 pandemic bring people together or push them apart? We predicted that perceived interdependence (measured as shared fate) would be associated with greater inclinations to cooperate, but that inclinations to cooperate would be compromised by COVID-19 infection risk. Inclinations to cooperate declined by a small margin over time as the pandemic unfolded. However, we saw substantial variation such that cooperation increased for some, remained stable for others, or decreased over time according to people's individual exposure to COVID-19 prevalence, dispositional perceptions of infection risk, and stable perceptions of interdependence with others. Overall, stable perceptions of interdependence with others were associated with higher and more stable, or increasing, inclinations to cooperate over time. In contrast, not only did COVID-19 infection risk interfere with people's perceived interdependence with others, but it was also associated with lower, or decreasing, inclinations to cooperate.

### How did inclinations to cooperate change during the COVID-19 pandemic?

Several studies report increasing cooperation following a crisis in the shorter term [3, 4, 47, 50–55]. However, the effect of crises on cooperation over the longer term can vary. Evidence for increasing cooperation over the longer term comes from some studies investigating intergroup conflict, in which cooperation is limited toward members of one's group [49, 73, 79].

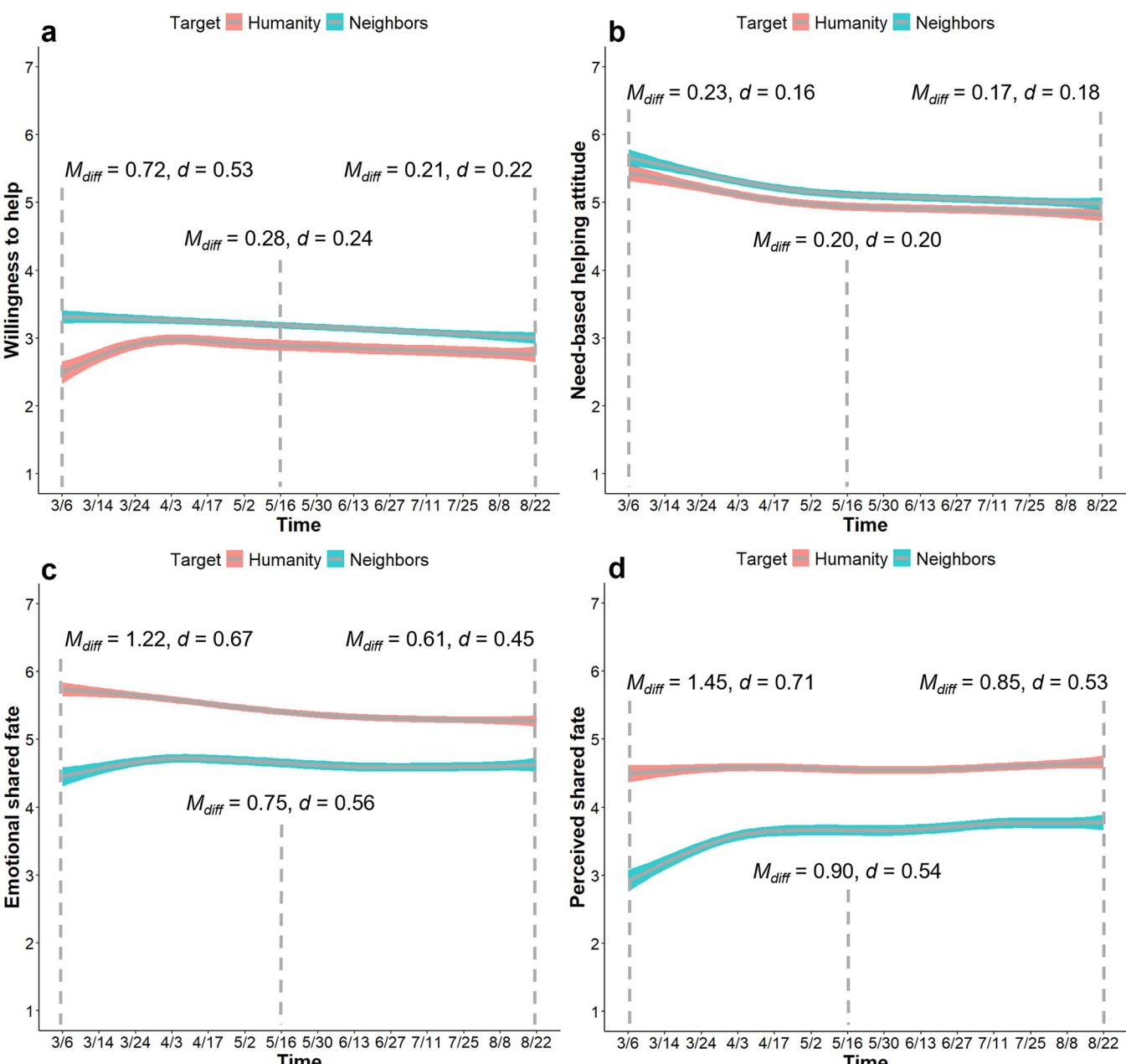

**Fig 9. Inclinations to cooperate and perceived interdependence between neighbors and humanity.** Participants reported greater willingness to help (a), and higher need-based helping attitude (b) toward neighbors (blue) than a person from a different country (red). Participants reported higher emotional shared fate (c) and perceived shared fate (d) with all of humanity (red) than with neighbors (blue). Need-based helping attitude at time 1 between neighbors and humanity $p = 0.005$, all other pairwise comparisons $p < 0.001$.

During disasters (e.g., floods and hurricanes), cooperation (measured as perceptions of received support) increases in the shorter term but declines in the longer term. This decline in perceived support is the culmination of high initial disaster impact, individuals experiencing chronically high need, and an increasing discrepancy between their perceived need for support and the support actually received [75–78]. Similarly, two studies during the COVID-19 pandemic among Spanish populations report that inclinations to cooperate decreased from March to August, 2020 [85, 86]. These findings that crises do not have uniform long-term impacts on

cooperation raise the question(s): how, for whom, and when does cooperation change over time?

## Cooperation was more stable or increased over time for people who perceived high interdependence with others

Our findings show that willingness to help neighbors and people from a different country by letting them move into your residence for a week declined sharply over time for people who perceived low and stable shared fate with neighbors and humanity. However, willingness to help did not decrease over time for people who perceived high and stable shared fate with others, instead staying higher and more stable. We also found that need-based helping attitude toward neighbors and people from a different country (i.e., agreeing that helping a person in need is the right thing to do) was higher at baseline and increased over time for people with high and stable emotional shared fate with neighbors and humanity. Need-based helping attitude toward people from a different country was lower at baseline and remained stable over time for people with low and stable emotional shared fate with humanity, while need-based helping attitude toward neighbors was both lower at baseline and decreased over time for people with low and stable emotional shared fate with neighbors.

While we found time × shared fate interactions on cooperation at the between-person level, we did not find any time × shared fate interactions on cooperation at the within-person level. These results indicate that within-person variation in shared fate arising from the pandemic (e.g., because of changes in COVID-19 prevalence, shared dysphoric experiences, opportunities to engage in mutual support, etc.) did not account for the changes in cooperation over time. Instead, the association between shared fate and changes in cooperation over time can be attributed to between-person differences in pre-existing shared fate with others. These findings may have implications for emergency preparedness and suggest that opportunities to build interdependence and feelings of shared fate within communities *prior* to an emergency could be important for sustaining, or even improving, cooperation when crises hit. Indeed, a longitudinal study during the COVID-19 pandemic finds that shared identity with one's community at baseline predicted future perceived support, but perceived support did not predict future shared identity [90]. However, more research is needed on the similarities and distinguishing features of different types of crises to avoid inappropriate overgeneralization of these results to other disaster settings. In crises where harm does not spread from person to person via contact (e.g., earthquakes), cooperation does not lead to a particularly heightened infection risk, and people may respond differently than they did in our study. Nevertheless, these results suggest that – for pandemics in particular – building feelings of interdependence in advance could help sustain cooperation during crises, even if they are long lasting as the COVID-19 pandemic has been.

## Inclinations to cooperate were compromised by infection risk

Based on existing research on the behavioral immune system [6, 7, 91], including recent investigations during the COVID-19 pandemic [70–72], we predicted that cooperation would be compromised by COVID-19 infection risk. Accordingly, willingness to help neighbors and a person from a different country decreased sharply over time for people who lived in areas with high and stable COVID-19 prevalence. However, for people who lived in areas with low and stable COVID-19 prevalence, willingness to help neighbors remained stable over time, while willingness to help a person from a different country decreased at a slower rate over time. We also found that willingness to help neighbors and a person from a different country decreased sharply over time when people experienced higher COVID-19 prevalence, but not when they

experienced lower COVID-19 prevalence. In addition, need-based helping attitude towards a person from a different country increased over time when people experienced lower COVID-19 prevalence, but not when they experienced higher COVID-19 prevalence. These findings indicate that people's inclinations to cooperate are adaptively calibrated to objective infection risk, such that they became less cooperative during time periods in which they were more likely to contract COVID-19.

Moreover, willingness to help and need-based helping attitude toward neighbors decreased over time at different rates depending on the individual's trait perceived infection risk. For people with low (compared to high) trait perceived infection risk, willingness to help neighbors decreased at a slower rate over time, while need-based helping attitude toward neighbors remained stable over time. Perceived infection risk was not associated with cooperation with a person from a different country. These findings indicate that within-person variation in perceived infection risk (e.g., because of changes in COVID-19 prevalence) did not account for the changes in cooperation over time. Instead, pre-existing between-person differences in trait perceived infection risk accounted for the decreasing cooperation with neighbors over time, but not for a person from a different country.

### How did perceived interdependence change during the COVID-19 pandemic?

**Infection risk interfered with increasing perceived interdependence.** While previous studies report long-term increases in measures strongly associated with perceived interdependence [74, 80–82], two studies during the COVID-19 pandemic reported decreases in self-other merging over time [85, 86]. As with inclinations to cooperate, here we saw substantial variation in perceived interdependence across and within individuals, increasing for some, remaining stable for others, or decreasing according to individual's exposure to COVID-19 prevalence. We found that, relative to their own location's overall COVID-19 prevalence, people reported increasing emotional and perceived shared fate with neighbors over time when they experienced lower COVID-19 prevalence, but not when they experienced higher COVID-19 prevalence. People also reported increasing emotional shared fate with humanity over time when they experienced lower COVID-19 prevalence. In contrast, emotional shared fate with humanity decreased by a small margin over time when people experienced higher COVID-19 prevalence.

These findings indicate that people's perceived interdependence with others adaptively calibrated to objective infection risk such that people perceived lower interdependence with others in occasions in which they were more likely to contract COVID-19. However, perceived infection risk was not associated with changes in perceived interdependence, suggesting that the effect of objective COVID-19 prevalence on changes in perceived interdependence operated through other cues and psychological mechanisms that are sensitive to such environmental changes (e.g., possibly disgust toward potentially infectious targets) [72]. In addition, our finding that perceived interdependence increased when people experienced lower objective COVID-19 infection risk suggests that people otherwise experienced cues associated with positive interdependence during the pandemic (e.g., shared dysphoric experiences, opportunities to engage in mutual support), but objective infection risk interfered with these positive interdependence cues (e.g., if gatherings were canceled due to high infection rates, the objective infection risk would remove some opportunities for experiencing cues associated with positive interdependence such as mutual support).

Taken together, our findings suggest that when people experienced lower COVID-19 prevalence, they reported increasing interdependence with others, in turn reporting more stable

willingness to help and increasing need-based helping attitudes over time. In contrast, experiencing higher prevalence of COVID-19 attenuated people's perceived interdependence with others, in turn decreasing willingness to help, and attenuating potential increases in need-based helping attitudes over time.

## Cooperating with the ingroup and scaling up interdependence with the outgroup?

People in the present study reported higher inclinations to cooperate with the ingroup (i.e., neighbors) than with the outgroup (i.e., a person from a different country). However, in contrast to our expectations, people reported higher emotional and perceived shared fate with all of humanity than with their neighbors to a strong degree, ranging from $d$ = 0.45 to 0.71. These results are surprising because previous work suggests that people care less about others as they become more distant [92–95] and as the number of others increases [96–98]. One potential interpretation of these results is that they may reflect an effort to scale up mutual support networks (e.g., investing in long-distance friendships) during large-scale crises, a pattern that has been observed across diverse societies [38, 99, 100]. For instance, perceptions of common fate following a flood were associated with greater shared group identity with those affected, which in turn was associated with greater expectations of support and shared goals [101]. We also see some evidence of this effect in the present study. Need-based helping attitude towards a person from a different country increased at more than twice the rate over time for people who reported high and stable emotional shared fate with humanity, compared to the increase in need-based helping attitude toward neighbors for people who reported high and stable emotional shared fate with neighbors.

An alternative explanation to consider here, hypothesized by Imada and Mifune [91] and empirically demonstrated by Ko et al. during the pandemic [72], is the possibility that people exert distance towards ingroup targets when such targets are believed to be sources of infection [72, 102–104]. Thus, rather than increasing shared fate towards more distant targets (i.e., all of humanity), under this alternative account our results might reflect decreasing shared fate with closer targets (i.e., neighbors). Our results are consistent with the idea that perceived infection risk can motivate avoidance toward ingroup targets when such targets are perceived to be sources of infection, but not with decreasing shared fate with ingroup members.

Consistent with the first hypothesis, we found that willingness to help neighbors decreased at a faster rate over time for people with high trait perceived infection risk than for people with low trait perceived infection risk, while need-based helping attitude toward neighbors decreased over time only for people with high trait perceived infection risk. Moreover, perceived infection risk was not associated with changes in cooperation with a person from a different country. Challenging the hypothesis that disease avoidance motivation led to decreasing shared fate with the ingroup, we found that people reported lower perceived interdependence with humanity, not neighbors, when they experienced higher COVID-19 prevalence.

## Limitations and future directions

One of the strengths of this study is that we measured inclinations to cooperate and perceived interdependence over many waves, allowing us to provide a detailed account of how these measures unfold over time during a pandemic. In addition, we sampled participants from 31 countries, allowing us to generalize findings to broader populations. At the same time, two limitations are that the study began on March 6, 2020, when COVID-19 was already underway for some period. Although we began data collection a few days before COVID was officially declared a pandemic, COVID-19-related concerns may have arisen for people before March 6,

2020. We cannot rule out the possibility that inclinations to cooperate and perceptions of interdependence were lower prior to the pandemic, increased prior to data collection, and we were only able to capture the changes that might have followed. Another limitation is that, while we sampled participants from many countries, the majority of participants came from Europe and North America, limiting our ability to generalize to other populations.

**Can disease prevalence be a source of positive interdependence?.**    Disease is commonly thought of as a source of *negative* interdependence. For example, when people feel disease-related disgust they perceive lower shared fate with others [5], become more prejudiced [64] and less cooperative [65], and engage in disparaging behaviors (e.g., damaging others' reputations) [105], all of which function to reduce interactions between people who have negative interdependence by way of infection risk. Our results are consistent with previous research at the within-person level: higher COVID-19 prevalence decreased inclinations to cooperate and interfered with positive perceived interdependence. However, one pattern of results at the between-person level violated our expectations. Perceived shared fate with neighbors increased over time for people who lived in areas with high and stable COVID-19 prevalence but remained stable over time for people who lived in areas with low and stable COVID-19 prevalence (Fig 8C). This suggests that people who lived in areas with high COVID-19 prevalence accurately perceived that they increasingly "rose and fell together" with neighbors as the pandemic wore on (i.e., because of the increasing number of infected individuals and the fact that infection risk goes up for everyone when local infection rates are higher).

What might account for these unexpected results? Situations involving diseases and the potential for disease transmission might generate positive interdependence because people in these settings might come together to help those who are sick or those who are isolated to avoid infection. Also, people may experience a need for and desire to engage in collective management of risk – in other words, working together with others to reduce the likelihood or severity of the threat. Because infectious disease spreads from person to person, collective and coordinated behavior to reduce the likelihood of that spread could benefit everyone, creating positive interdependence. Consistent with this interpretation, others have found that compared to those who did not engage in social distancing, those who engaged in social distancing to protect others during the first week of the pandemic reported greater self-other merging with others who engaged in social distancing [85]. Our results showing that perceived shared fate with neighbors increased over time for people who lived in areas with high COVID-19 prevalence indeed points to exactly this occurring during the pandemic. Further research should investigate the cues and situations under which disease prevalence might bring people together through perceptions of shared fate as we saw in our data, as opposed to pushing people apart.

**Did the pandemic change the attributions people made of cooperation and shared fate?.**    Another potential limitation of the present study is that our measures of inclinations to cooperate and perceived interdependence could have taken on different meanings during the pandemic compared to non-pandemic times. During non-pandemic times, perceptions of shared fate strongly predict higher cooperation across relationships [32]. Helping during pandemics is different from helping during other crises because helping in person can increase the risk of infection. Hence, our measure of willingness to help by letting others move into your home for a week might not have captured cooperation alone but also a tradeoff between the motivation to cooperate and people's willingness to incur the risk of infection [68, 106, 107]. Similarly, our measure of perceived shared fate with neighbors might not have captured only perceptions of positive interdependence but also perceptions of negative interdependence. This alternative interpretation might account for the unexpected findings that perceived shared fate with neighbors increased over time for people who lived in areas with high

COVID-19 prevalence, and for the finding that need-based helping attitude towards humanity increased over time for people who perceived low, not high, shared fate with humanity.

Finally, our measure of shared fate was developed to capture perceptions of shared fate towards specific kinds of relationships (e.g., acquaintances, cousins, friends, siblings). And, while situational factors such as infection risk during shared meals are associated with temporarily lower emotional shared fate [5], shared fate with others at the level of relationships might be generally stable over time [32]. Hence, other measures of situational, rather than relational, perceived interdependence might have better captured variation in people's fluctuating interdependence with others during the pandemic. While we recognize the difficulty of replicating findings due to the stochastic nature of pandemics, future studies should consider employing measures of cooperation that do not involve willingness to incur infection risk and measures of both relational and situational perceived interdependence [108, 109].

## Conclusion

We found that people came together during the COVID-19 pandemic, while also being pushed apart in some ways. Building on two theoretical frameworks, the behavioral immune system and fitness interdependence, we predicted that perceived interdependence would be associated with greater cooperation, but that cooperation would be compromised by infection risk. Overall, our findings suggest that people responded adaptively to the COVID-19 pandemic, reporting higher perceived interdependence with others and more stable or higher inclinations to cooperate when experiencing lower COVID-19 prevalence. When experiencing higher COVID-19 prevalence, people instead reported lower perceived interdependence with others and lower or attenuated inclinations to cooperate. However, while the pandemic pushed some people apart in terms of declining willingness to cooperate, perceived interdependence buffered against this decline in cooperation as the pandemic wore on. Despite a rising number of infected individuals, people with high and stable shared fate with others showed higher and more stable willingness to help over time, as well as increasing need-based helping attitude toward a person from a different country over time. These results show the power of people's sense of interdependence with others to create stability and resilience in their inclinations to cooperate, even during unexpectedly long-lasting crises.

## Supporting information

**S1 File. Supplemental method and analyses.**
(DOCX)

## Acknowledgments

We are grateful for the feedback we received from members of The Human Generosity Project (https://www.humangenerosity.org) and Daniel McNeish for sharing resources regarding pooled variance in multi-level modelling.

## Author Contributions

**Conceptualization:** Peter M. Todd, Athena Aktipis.

**Data curation:** Diego Guevara Beltran, Jessica D. Ayers, Hector Hurmuz-Sklias, Andrew Van Horn.

**Formal analysis:** Diego Guevara Beltran, Scott Claessens.

**Funding acquisition:** Joe Alcock, Lee Cronk, Peter M. Todd, Athena Aktipis.

**Investigation:** Diego Guevara Beltran, Jessica D. Ayers, Athena Aktipis.

**Methodology:** Diego Guevara Beltran, Jessica D. Ayers, Athena Aktipis.

**Project administration:** Diego Guevara Beltran, Jessica D. Ayers, Cristina Baciu, Nicole M. Hudson, Andrew Van Horn, Peter M. Todd, Athena Aktipis.

**Supervision:** Joe Alcock, Lee Cronk, Peter M. Todd, Athena Aktipis.

**Visualization:** Diego Guevara Beltran, Scott Claessens, Hector Hurmuz-Sklias.

**Writing – original draft:** Diego Guevara Beltran, Jessica D. Ayers, Peter M. Todd, Athena Aktipis.

**Writing – review & editing:** Diego Guevara Beltran, Jessica D. Ayers, Scott Claessens, Joe Alcock, Cristina Baciu, Lee Cronk, Nicole M. Hudson, Hector Hurmuz-Sklias, Geoffrey Miller, Keith Tidball, Andrew Van Horn, Pamela Winfrey, Emily Zarka, Peter M. Todd, Athena Aktipis.

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
