## [Decision Letter · Decision Letter 0]

15 Feb 2024

PONE-D-23-42608Shared Fate was Associated with Sustained Cooperation During the COVID-19 PandemicPLOS ONE

Dear Dr. Guevara Beltran,

Thank you for submitting your manuscript to PLOS ONE. After careful consideration, we feel that it has merit but does not fully meet PLOS ONE’s publication criteria as it currently stands. Therefore, we invite you to submit a revised version of the manuscript that addresses the points raised during the review process.

Editor's comments:

Two reviewers familiar with research on the effects of natural or social threat on psychology have read your manuscript.

Both reviewers recognize the importance of your research using a large sample and longitudinal study, but are concerned about the lack of explanation in the manuscript. Reviewer 1 also noted deficiencies in several references. Reviewer 2 made some suggestions for the analysis.

Please review the reviewers' comments and revise your manuscript according to their point-by-point responses.

I would also like to point out some minor issues. 

Regarding the description of Figure 4(c), I think there seems to be a discrepancy. Please check it. 

Also, please review the submission guidelines carefully and correct any errors. Reference style deviates from the guidelines.

https://journals.plos.org/plosone/s/submission-guidelines#loc-references

We look forward to receiving your revised manuscript.

Kind regards,

Yutaka Horita

Academic Editor

PLOS ONE

Journal Requirements:

3.We note that the grant information you provided in the ‘Funding Information’ and ‘Financial Disclosure’ sections do not match. 

"This study was funded by the Interdisciplinary Cooperation Initiative, the President’s Office of Arizona State University, the Cooperation Science Network, the Institute for Mental Health Research, the University of New Mexico, the Indiana University College of Arts & Sciences, the Rutgers University Center for Human Evolutionary Studies, the Charles Koch Foundation, the John Templeton Foundation, and the President’s Office of the University of Arizona. Any opinions, findings, conclusions or recommendations expressed in this material are those of the authors and do not necessarily reflect the views of the funding entities mentioned above."

"We have no conflicts of interest to disclose."

7. We note that [Figure 1] in your submission contain [map/satellite] images which may be copyrighted. All PLOS content is published under the Creative Commons Attribution License (CC BY 4.0), which means that the manuscript, images, and Supporting Information files will be freely available online, and any third party is permitted to access, download, copy, distribute, and use these materials in any way, even commercially, with proper attribution. For these reasons, we cannot publish previously copyrighted maps or satellite images created using proprietary data, such as Google software (Google Maps, Street View, and Earth). For more information, see our copyright guidelines: http://journals.plos.org/plosone/s/licenses-and-copyright.

a. You may seek permission from the original copyright holder of Figure [1] to publish the content specifically under the CC BY 4.0 license.  

Reviewers' comments:

Reviewer's Responses to Questions

**Comments to the Author**

1. Is the manuscript technically sound, and do the data support the conclusions?

Reviewer #1: Yes

Reviewer #2: Yes

2. Has the statistical analysis been performed appropriately and rigorously? 

Reviewer #1: I Don't Know

Reviewer #2: Yes

3. Have the authors made all data underlying the findings in their manuscript fully available?

Reviewer #1: Yes

Reviewer #2: Yes

4. Is the manuscript presented in an intelligible fashion and written in standard English?

Reviewer #1: Yes

Reviewer #2: Yes

5. Review Comments to the Author

Reviewer #1: This study looks at the relationship between perceived interdependence and reports and intentions to provide social support over time during the first 5 months of the pandemic. There is a very impressive dataset using a panel study with multiple waves and an international sample. The paper is clearly written and the rationale is developed logically. As a social psychologist for me there were some gaps in what was otherwise what looked like an up to date and comprehensive literature review – the introduction seemed to be based more on anthropology and evolutionary psychology than social psychology, despite the relevance of the latter to the research questions. I can’t comment on all the statistics described – the editors need to get the views of a specialist on these. The conclusions followed logically from the results and there was some awareness of the limitations of the study (though see below).

My main concern is with the measures used. The study uses single-item measures, with no indication of their origin, reliability or validity. I assume that this decision was taken to make the questionnaire more acceptable to respondents? The items seem intuitive so they are probably mostly ok (the authors themselves acknowledge that one of the measures of social support is seriously ambiguous), but I would have expected some explanation and justification of them in the method section and an explicit acknowledgment of the limitations of them in the discussion.

The other issue is I was slightly surprised that there was no account of what was happening with the pandemic in the period in question. Surely participant answers on questions concerning common fate and helping will be affected by whether there is a wave. (We know for example that there was a large increase in helping behaviour in response to the first wave.) In the periods between waves, when politicians often played down the pandemic and attempted to open society up again, there would have been less perceived need for help.

My other comments are questions and suggestions for the authors to consider.

Introduction:

p. 3/11 of the pdf ‘Sometimes

67 people become fearful and aggressive, stealing resources and committing crimes (Drury et al.,

68 2013)’

This is an odd choice of reference given that the paper cited is about disaster myths.

p. 3/11 of the pdf

‘Crises can create positive interdependence, eliciting perceptions of shared fate and

70 promoting cooperation (e.g., Andrighetto et al., 2016; Drury et al., 2016)’

The Drury et al. study found no direct relation between common fate and cooperation. In line with the social identity approach, common fate predicted shared identity and it was shared identity that predicted cooperation.

p. 4/ 12 of the pdf

‘and panic buying were prevalent aspects of

94 behavior (The White House, 2020)’

I don’t know any source suggesting that so called panic buying was ‘prevalent’ during the pandemic. The excellent study by Kantar showed that that just a small number of people bought extra, and that only a minority of these bought very large amounts. Moreover these episodes of extra buying occurred at the start of the pandemic rather than operating throughout 2020. The source cited (The White House) was ‘page not found’ so I couldn’t check what it says.

p. 8/16 of the pdf

‘After a natural disaster (i.e., flood,

220 earthquake)’

Some argue that the term ‘natural disasters’ is misleading: see https://www.nonaturaldisasters.com/

p. 18 of the pdf

‘For example, cooperation could marginally increase over time, as has been reported in

267 most crises.’

This claim is not referenced. Also, in the readings I’m more aware of, the opposite is the case. Immediately after a disaster there is extensive cooperation, which then declines afterwards as people run out of resources. Krys Kaniasty has written extensively about this and is probably worth including. For application of his approach to cooperation during Covid (and for literature and data on decline in social support over the first year of the pandemic, see:

Ntontis, E., Fernandes-Jesus, M., Mao, G., Dines, T., Kane, J., Karakaya, J., Perach, R., Cocking, C., McTague, M., Schwarz, A., Semlyen & J. & Drury, J. (2022). Tracking the nature and trajectory of social support in Facebook mutual aid groups during the COVID-19 pandemic. International Journal of Disaster Risk Reduction. https://doi.org/10.1016/j.ijdrr.2022.103043

Methods:

Why this sample size?

Why the extra recruitment at time 3?

Why these particular timepoints?

I can’t follow how the t1-2 data can be used with the other data.

Demographics – provide age range, not just SD?

How long did it take and how much were people paid?

Discussion:

p. 31 of the pdf

‘people high on

589 self-other-merging, a marker of perceived interdependence’

But there wasn’t a measure of ‘self other merging’, was there?

p. 31 of the pdf

‘Our results have potential implications for emergency preparedness and suggest that

609 opportunities to build interdependence within communities prior to an emergency might be

610 important for sustaining cooperation when crises hit.’

This is the basis of many countries’ approach to community resilience, so cite some examples.

Why is table 3 in the discussion?

Reviewer #2: Dear Authors,

This study observed through a longitudinal study conflicting psychological changes in the COVID-19 pandemic: whether people's cooperative behavior was promoted or inhibited. The study captures continuous responses through a longitudinal survey, providing valuable insights into the development of psychological responses during a public health crisis. However, some areas could be improved to enhance clarity and elaboration.

1. Introduction

1-1. Line 161: Request for additions regarding the characteristics of the COVID-19 pandemic

Unlike its other crises, the COVID-19 pandemic did not initially have robust guidelines for crisis management or clear-cut right answers for cooperative behavior. Also, unlike other crises that were localized in time and geography, this pandemic was a dynamic, ongoing global crisis. I ask that you consider adding that this was a dynamic, highly unknowable, and endlessly stressful situation in the manuscript. In this connection, please note that the comparison with the terrorist incidents (i.e., localized crises) cannot be made simply (Lines 191 to 200).

Also, under such an unknown and ongoing crisis, some people would have acted as supporters or selfishly according to "social norms" such as government guidelines and local community rules. Simple dichotomous statements such as supportive and selfish people (Lines 85 to 100) risk oversimplifying the discussion.

1-2. Lines 140-160: Request for causal clarification of arguments based on fitness interdependence theory

This paragraph suggests that fitness interdependence theory may go beyond interrelationships among individuals and within relatively small groups to the larger group level. If my understanding is correct, why not cite and organize one of the clues, "a history of sharing" (Line 122), to make it clearer to the reader?

I was under the impression that the statement in this paragraph is the reverse of the causal relationship with the fitness interdependence theory. That is, in the fitness interdependence theory, I understood that the three clues in line 114 exist as causes, which in turn affect each other in terms of disaster survival (lines 133 to 139). However, from lines 140 to 160, it seemed to me that there is another process of acquiring an interdependent perception of the experience of disasters such as earthquakes and terrorist attacks as cues.

2. Analysis

2-1. Line 346: Request for clarification regarding cluster-mean score

How was the individual baseline handled when calculating the cluster-mean scores?

2-2.Suggestions for analysis model

Please consider including the number of infections per day in the participant's country at each time point in the analysis model.

It is possible that regional differences in the perceived infection risk at a given point in time, such as whether the infection situation in a country is severe or not, may arise.

3. Results

3-1: Suggestions for modification of the description

For results with small effects, I request careful description to prevent readers from over-interpreting them (e.g., Lines 394-395, 405-406). Also, please consider presenting a visual figure showing the 95% confidence interval.

4. Discussion

4-1. Line 580: Request for additional prior studies

Imada & Mifune (2021) suggest that when an infectious disease is prevalent in a group, the behavior of avoiding other members of the inner group may increase (data not verified).

Please consider bolstering your argument through the following and other papers.

Imada, H., & Mifune, N. (2021). Pathogen threat and in-group cooperation. Frontiers in Psychology, 12, 2568. https://doi.org/10.3389/fpsyg.2021.678188

6. PLOS authors have the option to publish the peer review history of their article (what does this mean?). If published, this will include your full peer review and any attached files.

Reviewer #1: **Yes: **John Drury

Reviewer #2: **Yes: **Mei Yamagata

---

## [Author Response · Author response to Decision Letter 0]

10 Jun 2024

Dear Dr. Horita, 

My coauthors and I are grateful for the opportunity to revise our manuscript, and submit it for reconsideration at PLOS ONE. We would like to thank you and the reviewers for taking the time to read our manuscript and provide such thorough, thoughtful, and constructive comments. We have read the reviews carefully, and have made substantial revisions addressing the concerns expressed. Most importantly, (1) we included a measure of COVID-19 prevalence and re-ran our analyses to include this additional covariate. (2) We addressed reviewers’ concerns regarding citations and nomenclature, (3) as well as reviewer’s concerns with sample characteristics and measures employed. We’re especially grateful with the reviewers’ suggestion to include cases of COVID-19 to our analysis. We are confident our paper now presents a more complete, nuanced, and interesting account of people’s inclinations to cooperate and perceived interdependence during the early period of the pandemic. 

Below we provide a consolidated point-by-point summary of reviewers’ concerns, explaining how (and where) we addressed them in the revised manuscript. To ease the readability of this document, editor and reviewer comments are shown in bold, our responses are shown below, and changes to the manuscript are shown indented and italicized. We are optimistic that our revisions have sufficiently addressed each of the reviewers’ concerns, questions, and suggestions, and we appreciate the ways in which the paper has improved as a result. We look forward to receiving your assessment of the revised manuscript in due time. 

Best,

Diego Guevara Beltran, PhD

Editor comments

1. Regarding the description of Figure 4(c), I think there seems to be a discrepancy. 

We thank the editor for noticing this oversight. We have redone our graphs and made sure the labels properly represent the statistical results.

2. Citations

We edited the in-text citations and reference section to Vancouver style.

3. Formatting

We edited the manuscript to comply with formatting requirements. 

REVIEWER 1

1. The study uses single-item measures, with no indication of their origin, reliability or validity. I assume that this decision was taken to make the questionnaire more acceptable to respondents? The items seem intuitive so they are probably mostly ok (the authors themselves acknowledge that one of the measures of social support is seriously ambiguous), but I would have expected some explanation and justification of them in the method section and an explicit acknowledgment of the limitations of them in the discussion.

We included the following text in the method section to address the reviewer’s concern about the items: 

Depending on the setting and source, fitness interdependence can manifest psychologically through perceptions of interdependence such as perceived similarity in the context of relatedness or kin relationships, via self-other-merging in the context of intergroup or large scale disasters; and relationship closeness in the context of romantic relationships. At their core, these psychological manifestations of fitness interdependence track the extent to which other’s outcomes are likely to influence one’s own outcomes. The shared fate scale was designed to measure perceptions of interdependence globally (i.e., across settings, sources, and relationships), and is a reliable measure comprised of two subscales that have shown concurrent, discriminant, convergent, and predictive validity (32). 

Perceived shared fate (e.g., [target] and I rise and fall together) tracks beliefs about the extent to which partners’ rewarding and aversive outcomes are likely to translate into a personal reward or loss. For example, extent to which people believe a partner’s job promotion will have a positive influence on one’s life, or extent to which people believe a partner’s parents’ death will have a negative influence on one’s life. Emotional shared fate (e.g., when [target] succeeds, I feel good) tracks affective responses to partners’ outcomes. Specifically, the intensity of emotional shared fate corresponds to the appraisals people make about the extent to which others’ positive and negative outcomes are likely to translate into a personal reward or loss. For example, how good one feels when one’s partner receives a job promotion, or how bad one feels when said partner’s parents pass away (32). 

Perceived and emotional shared fate inform people about the extent to which they are positively interdependent with specific relationship partners, in turn predicting how much to value such partners, and how much to invest in their welfare (32,35). Although the shared fate scale was developed as a six-item measure, here we employed only one item per subscale to reduce participant fatigue, and maximize the allocation of time and resources for other survey items. We selected the item “when [target] succeeds, I feel good” as the index of emotional shared fate because at Time 1, this item showed the strongest correlation with need-based helping attitude towards neighbors (r = 0.35, p < 0.001), compared to other items (r’s = 0.07 to 0.28). Similarly, we selected the item “[target] and I rise and fall together” as the index of perceived shared fate because at Time 1, this item showed the strongest correlation with willingness to help a person from a different country (r = 0.24, p < 0.001), compared to the other items (r’s = 0.01 to 0.22).

In the limitations section, we discuss limitations of our measures, including that the willingness to help item could have indexed a tradeoff between disease avoidance and willingness to help; as well as limitations regarding our shared fate items, including the fact that the shared fate scale was designed to measure interdependence at the level of relationships, rather than situations, and that perceived shared fate could have indexed both perceptions of negative and positive interdependence.

2. The other issue is I was slightly surprised that there was no account of what was happening with the pandemic in the period in question. Surely participant answers on questions concerning common fate and helping will be affected by whether there is a wave. 

We thank the reviewer for making this excellent suggestion, we reanalyzed the data including incidence of COVID-19 infections. We report updated analyses in the main text and supplements. We also include this text in the method section: 

To determine the local prevalence of COVID-19, we matched participants’ location information to the COVID-19 Data Repository maintained by the Center for Systems Science and Engineering (CSSE) at Johns Hopkins University (9). Using this database, we used the latitude and longitude of each participants’ postal code and matched the postal codes to the corresponding location in the COVID-19 Data repository to determine cumulative incidence rates of COVID-19 (i.e., confirmed cases per 100,000 individuals) of 520 participants. We were not able to match the prevalence of COVID-19 for participants who did not provide postal codes or provided unidentifiable postal codes.

3. p. 3/11 of the pdf ‘Sometimes people become fearful and aggressive, stealing resources and committing crimes (Drury et al., 2013). This is an odd choice of reference given that the paper cited is about disaster myths.

We apologize for this oversight, we have removed this citation and replaced it for a paper showing both decreases (at the local county level), and increases (in the surrounding areas) in criminal activity following disasters (1). 

4. p. 3/11 of the pdf ‘Crises can create positive interdependence, eliciting perceptions of shared fate and promoting cooperation (e.g., Andrighetto et al., 2016; Drury et al., 2016)’ The Drury et al. study found no direct relation between common fate and cooperation. In line with the social identity approach, common fate predicted shared identity and it was shared identity that predicted cooperation.

We edited this sentence: 

Crises can create positive interdependence, eliciting perceptions such as shared fate and shared identity, and in turn promoting cooperation (3,4).

5. p. 4/ 12 of the pdf ‘and panic buying were prevalent aspects of behavior (The White House, 2020)’. I don’t know any source suggesting that so called panic buying was ‘prevalent’ during the pandemic. The excellent study by Kantar showed that just a small number of people bought extra, and that only a minority of these bought very large amounts. Moreover these episodes of extra buying occurred at the start of the pandemic rather than operating throughout 2020. The source cited (The White House) was ‘page not found’ so I couldn’t check what it says.

We have corrected this sentence: 

At the same time, more people were purchasing household goods (e.g., cleaning supplies, non-perishable foods) during March 2020 compared to the previous year (18), fueling the belief that people were looking out for themselves (e.g., hoarding) and incentivizing others to behave similarly (19). Such fears, while factually unfounded (18), even led the secretary of the United States to take action to mitigate the potential risk of hoarding and price gouging (20).

The executive order referenced above can be found here: https://trumpwhitehouse.archives.gov/presidential-actions/executive-order-preventing-hoarding-health-medical-resources-respond-spread-covid-19/

6. p. 8/16 of the pdf ‘After a natural disaster (i.e., flood, earthquake)’ Some argue that the term ‘natural disasters’ is misleading: see https://www.nonaturaldisasters.com/

We deleted “natural” in all mentions of disasters.

7. p. 18 of the pdf ‘For example, cooperation could marginally increase over time, as has been reported in most crises.’ This claim is not referenced. Also, in the readings I’m more aware of the opposite is the case. Immediately after a disaster there is extensive cooperation, which then declines afterwards as people run out of resources. Krys Kaniasty has written extensively about this and is probably worth including. For application of his approach to cooperation during Covid (and for literature and data on decline in social support over the first year of the pandemic, see: Ntontis, E., Fernandes-Jesus, M., Mao, G., Dines, T., Kane, J., Karakaya, J., Perach, R., Cocking, C., McTague, M., Schwarz, A., Semlyen & J. & Drury, J. (2022). Tracking the nature and trajectory of social support in Facebook mutual aid groups during the COVID-19 pandemic. International Journal of Disaster Risk Reduction. https://doi.org/10.1016/j.ijdrr.2022.103043

We thank the reviewer for pointing us in the direction of Kaniasty. His work indeed is very relevant. We included the following text citing his research:

In parallel to volunteer rates following terrorist attacks, people who experienced greater losses during a flood received help shortly after the flood, but perceptions of received support declined considerably six months after the flood (75,76). This decline in support following crises is a result of both exposure to crises, and unmet needs or expectations, rather than objective declines in received support per se. Controlling for received support, people who experienced more losses during a hurricane reported lower perceptions of received support two years after the hurricane, suggesting that unmet needs contribute to declines in perceived support regardless of objectively received support (77). Moreover, greater exposure to disasters can onset or exacerbate mental health problems (e.g., PTSD), which in the longer term (i.e., 12-18 months post disaster), can lead to steeper declines in perceptions of received support (78).

We also include the following text citing Ntontis et al: 

Studies during the COVID-19 pandemic point to a pattern of initial increases in cooperation, followed by decreases in the longer term. For example, activity in online mutual aid groups (i.e., posts requesting and offering help) increased sharply during the first wave of the pandemic (i.e., March 2020), but declined to baseline levels by July 2020 (84).

8. Why this sample size?

We were under time pressure to launch the survey (1 week) given access to research funds available at the time, as well as the need to launch the survey before plausible changes continued to accumulate. Given this time pressure, we relied on recommendations to detect two- and three-way interactions in longitudinal mixed-effects models. With 400 participants and a minimum of 8 time periods, we can confidently (i.e., with 80% power) detect time x covariate interactions of even small effect sizes. With 300-500 participants, we can also detect with confidence medium-to-large three-way interactions (He & Leon, 2010; see link below). Thus, we aimed to recruit 500 participants. https://www.tandfonline.com/doi/full/10.1080/10543401003618819

9. Why the extra recruitment at time 3?

With 20% attrition at Time 2, we decided to recruit another set of participants in hopes of obtaining 500 participants by the end of the survey. 

10. Why these particular timepoints?

We sought to maximize the number of time points, given available funds, so that we could get a detailed account of the developmental trajectory of measures employed. Maximizing time points allowed us to (a) document the within-person variability in perceived interdependence and inclinations to cooperate over time, and (b) measure perceived infection risk and perceived interdependence as a time-varying covariate (see Figures 2-3).

11. I can’t follow how the t1-2 data can be used with the other data.

An advantage of mixed-effects models is the maximum likelihood estimation method. This allows the estimation of parameters even with missing data at varying time points. In addition, estimates/parameters (e.g., slope of time), are informed by all available information at level-1 (i.e., within-person observations). However, it is weighed by the information provided at the level of the cluster (i.e., the between-person level). The model can make a “best guess” of a person’s individual trajectory/slope based on all the pooled information. Just like in single-level regression, the model finds the line of best fit when looking across all individual slopes/trajectories (i.e., the fixed effects, such as the average slope of all individual slopes, or the average intercept of all individual intercepts), but then uses this information in combination with the random effects (i.e., estimates of between-person variance surrounding fixed-effects) to also make guesses/predictions about the person-specific estimates (e.g., individual-level intercepts, slopes). Below is a graphical representation of this process (see participant 373, 374 for cases with missing data; green line shows the predicted individual trajectory when “pooling” information from all individual trajectories): https://www.tjmahr.com/plotting-partial-pooling-in-mixed-effects-models/

12. Demographics – provide age range, not just SD?

We edited the reporting of age:

(Mage = 28.65, SDage = 10.23, Minage = 18, Maxage = 80, 50.35% men).

13. How long did it take and how much were people paid?

Duration per time period is reported on Table 1. We now also add the following sentence:

Participants were compensated with $1.75 USD for Times 1-4 ($10.10 USD/hour on average), and an average of $1.50 USD ($8.64 USD/hour on average) from times 5-14.

14. p. 31 of the pdf ‘people high on self-other-merging, a marker of perceived interdependence’ But there wasn’t a measure of ‘self other merging’, was there?

We apologize for the oversight, we used self-other-merging for identity fusion. We deleted the referenced sentence from the revised manuscript.

15. p. 31 of the pdf ‘Our results have potential implications for emergency preparedness and suggest that opportunities to build interdependence within communities prior to an emergency might be important for sustaining cooperation when crises hit.’ This is the basis of many countries’ approach to community resilience, so cite some examples.

We added the following text, citing Ntonis et al (2020): 

Indeed, a longitudinal study during the COVID-19 pandemic finds that shared identity with one’s community at baseline pre

---

## [Decision Letter · Decision Letter 1]

12 Jul 2024

Shared fate was associated with sustained cooperation during the COVID-19 pandemic

PONE-D-23-42608R1

Dear Dr. Guevara Beltran,

We’re pleased to inform you that your manuscript has been judged scientifically suitable for publication and will be formally accepted for publication once it meets all outstanding technical requirements.

Kind regards,

Yutaka Horita

Academic Editor

PLOS ONE

Additional Editor Comments (optional):

Thank you for submitting the revision. Unfortunately, one reviewer declined to review the manuscript again for personal reasons, but I have reviewed the manuscript instead to ensure that your revisions address their comments. I think that there are no major problems with both content and format, and the paper meets the criteria for publication.

Reviewers' comments:

Reviewer's Responses to Questions

**Comments to the Author**

1. If the authors have adequately addressed your comments raised in a previous round of review and you feel that this manuscript is now acceptable for publication, you may indicate that here to bypass the “Comments to the Author” section, enter your conflict of interest statement in the “Confidential to Editor” section, and submit your "Accept" recommendation.

Reviewer #2: All comments have been addressed

2. Is the manuscript technically sound, and do the data support the conclusions?

Reviewer #2: Yes

3. Has the statistical analysis been performed appropriately and rigorously? 

Reviewer #2: Yes

4. Have the authors made all data underlying the findings in their manuscript fully available?

Reviewer #2: Yes

5. Is the manuscript presented in an intelligible fashion and written in standard English?

Reviewer #2: Yes

6. Review Comments to the Author

Reviewer #2: Thank you for addressing the raised comments appropriately and positively. The reanalysis of the data, including regional COVID-19 prevalence, and the subsequent discussion are particularly interesting. Additionally, the structure of the introduction has been organized, significantly strengthening the basis of the discussion. The characteristics of the COVID-19 pandemic have been thoroughly considered, clearly distinguishing it from other crises.

I apologize for any confusion caused by my inadequate explanation regarding the cluster-mean scores. My concern was about the potential influence of the individual baseline (initial time point scores) on the subsequent time series patterns. One way to address this is by including the baseline scores as covariates in the model. However, after re-reading the authors' comments and the manuscript, I understand that the mean scores were calculated using the entire time series data for each individual and that within-person effects were considered. This approach does not rely solely on specific time point scores and does not compromise the results of this study. Therefore, I believe this is not an issue.

I am confident that this paper meets the necessary standards for publication.

7. PLOS authors have the option to publish the peer review history of their article (what does this mean?). If published, this will include your full peer review and any attached files.

Reviewer #2: **Yes: **Mei Yamagata

---

## [Editor Report · Acceptance letter]

8 Aug 2024

PONE-D-23-42608R1 

PLOS ONE

Dear Dr. Guevara Beltran, 

I'm pleased to inform you that your manuscript has been deemed suitable for publication in PLOS ONE. Congratulations! Your manuscript is now being handed over to our production team.

Kind regards, 

on behalf of

Dr. Yutaka Horita 

Academic Editor

PLOS ONE